# Parallel reduction in flowering time from de novo mutations enable evolutionary rescue in colonizing lineages

Andrea Fulgione[1,2,3,9], Célia Neto [1,9], Ahmed F. Elfarargi [1], Emmanuel Tergemina [1], Shifa Ansari[1], Mehmet Göktay[1], Herculano Dinis[4,5], Nina Döring[1], Pádraic J. Flood[1], Sofia Rodriguez-Pacheco[1], Nora Walden [6,7], Marcus A. Koch [6], Fabrice Roux[8], Joachim Hermisson[2] & Angela M. Hancock [1,2✉]

Understanding how populations adapt to abrupt environmental change is necessary to predict responses to future challenges, but identifying specific adaptive variants, quantifying their responses to selection and reconstructing their detailed histories is challenging in natural populations. Here, we use *Arabidopsis* from the Cape Verde Islands as a model to investigate the mechanisms of adaptation after a sudden shift to a more arid climate. We find genome-wide evidence of adaptation after a multivariate change in selection pressures. In particular, time to flowering is reduced in parallel across islands, substantially increasing fitness. This change is mediated by convergent de novo loss of function of two core flowering time genes: *FRI* on one island and *FLC* on the other. Evolutionary reconstructions reveal a case where expansion of the new populations coincided with the emergence and proliferation of these variants, consistent with models of rapid adaptation and evolutionary rescue.

[1] Max Planck Institute for Plant Breeding Research, Cologne, Germany. [2] Mathematics and Bioscience, Department of Mathematics and Max F. Perutz Labs, University of Vienna, Vienna, Austria. [3] Vienna Graduate School for Population Genetics, Vienna, Austria. [4] Parque Natural do Fogo, Direção Nacional do Ambiente, Praia, Santiago, Cabo Verde. [5] Associação Projecto Vitó, São Filipe, Fogo, Cabo Verde. [6] Centre for Organismal Studies (COS) Heidelberg, Biodiversity and Plant Systematics, Heidelberg University, Heidelberg, Germany. [7] Biosystematics, Wageningen University, Wageningen, The Netherlands. [8] LIPME, Université de Toulouse, INRAE, CNRS, Castanet-Tolosan, France. [9] These authors contributed equally: Andrea Fulgione, Célia Neto. ✉email: hancock@mpipz.mpg.de

One in eight of the world's existing plant and animal species are at risk of extinction due to human-mediated environmental change[1]. To forecast and mitigate risk, it is necessary that we understand the mechanisms of adaptation to novel environmental challenges. On the one extreme, adaptation can be highly polygenic, with contributions from many small effect variants[2–5]. Conversely, when selection pressures are very strong and existing genetic variation is low, large-effect variants are expected to provide a crucial contribution to adaptation[6–8]. Theoretical models show the importance of genetic diversity and the strength of selection for shaping the architecture of adaptive response[6,7,9–14].

In practice, reconstructing detailed adaptive histories in natural populations is challenging. However, long-range colonization events can represent powerful natural experiments where populations are deposited in replicate in a new environment[9,15–19]. The resulting isolated populations provide an opportunity to examine evolutionary processes in the absence of confounding from admixture and secondary contact.

A single *Arabidopsis* line from Cape Verde (Cvi-0) was collected 37 years ago[20] and has since been studied extensively both at the phenotypic and genetic levels. This accession has been an enigma because it lies geographically and climatically far outside of the core range of *Arabidopsis*. The Cape Verde Islands (CVI) archipelago consists of ten islands located between 14.80 and 17.20 degrees north of the equator and 570 km from the coast of Senegal. The flora in CVI is a mix of native species that reached the islands via long-range dispersal from mainland Africa and Macaronesia and species introduced since 1456, when humans first settled in CVI[21,22]. Precipitation in CVI is limited and unpredictable—so that plants must grow quickly and reproduce in the short time when water is available[21]. The wealth of information for Cvi-0 together with the isolation of *Arabidopsis* in CVI provided a potentially powerful case to connect the genetic basis of adaptive change with ecological drivers and fitness differentials.

Here, we sequence the genomes of 335 *Arabidopsis* lines from CVI and use a combination of population genetic inference and trait-mapping to reconstruct their evolutionary history. In small colonizing populations, the strength of genetic drift is strong[14]. However, in CVI *Arabidopsis*, where the colonizing population faced strong selection pressures, we find genome-wide signatures of adaptive evolution and show that parallel reduction in flowering time was a crucial first adaptive step. We identify functional variants responsible for an approximately 30-day reduction in flowering time and show these had a large selective advantage, consistent with expectations under the Fisher-Orr model of adaptation[23,24]. Finally, we discuss the relevance of our findings to observations in continental populations of *A. thaliana* and across species.

## Results

### Reconstructing demographic history of CVI *Arabidopsis* from genome-wide patterns of variation. We collected *Arabidopsis* across its distribution in CVI (Fig. 1a, Supplementary Fig. 1, Supplementary Data 1), where it is limited to the islands Santo Antão and Fogo, and sequenced complete genomes of 335 lines. Compared to Eurasian and Moroccan collection locations, the *Arabidopsis* habitat in Cape Verde is more arid (median aridity index in CVI: 0.21, Morocco: 0.25, Eurasia: 0.78; Mann–Whitney–Wilcoxon (MWW) for CVI-Eurasia: $p = 3.41 \times 10^{-35}$ and CVI-Morocco: $p = 5.97 \times 10^{-4}$) with higher precipitation seasonality (median in CVI: 144.24, Morocco: 54.00, Eurasia: 25.94; MWW CVI-Eurasia: $p = 2.01 \times 10^{-36}$ and CVI-Morocco: $p = 3.8 \times 10^{-11}$), and a shorter growing season (median in CVI: 3.5 months, Morocco: 8 months, Eurasia: 8 months; MWW CVI-Eurasia: $p = 2.72 \times 10^{-35}$ and CVI-Morocco: $p = 4.13 \times 10^{-12}$) (Supplementary Fig. 2, Supplementary

Data 2). The strong climatic divergence of CVI suggests nascent CVI populations may have been subject to strong selection.

We reconstructed the colonization history of CVI *Arabidopsis* by analysing CVI genomes together with published data[25,26]. Genome-wide, the two Cape Verde islands cluster tightly together and are nested within the Moroccan clade (Fig. 1b). Diversity within islands is 73.3- and 62.3-fold reduced compared to the continent ($\theta_W$ (Santo Antão) $= 7.59 \times 10^{-5}$, $\theta_W$ (Fogo) $= 8.93 \times 10^{-5}$, $\theta_W$ (Morocco) $= 5.56 \times 10^{-3}$; Supplementary Table 1) and there is almost no shared variation between the islands and Morocco or between the two Cape Verde Islands (Fig. 2a, b). Genome-wide, 99.9% of variants in CVI are absent in Morocco and 99.4% of variants segregating in Cape Verde are private to a single island. Similarly, at 4-fold degenerate sites, 99.9% are private to Cape Verde and 98.2% are private to only one island (Fig. 2b). Linkage disequilibrium decays rather rapidly in each island population (Supplementary Fig. 3), consistent with the near-complete loss of segregating variation with colonization (i.e., lack of deep population structure) and subsequent population expansion[27,28].

These levels of differentiation between CVI and the Moroccan mainland as well as between CVI islands are striking. Divergence is higher than that observed between species pairs in the *Arabidopsis* genus, which ranges from 72.6% to 96.9% private 4-fold degenerate segregating variants[29]. As a result, each Cape Verde island population forms a diverged, monophyletic group and is thus phylogenetically distinct, and will be treated as such here for the purposes of genetic analysis. Further, the patterns we observe for these lineages are analogous to those inferred for most named endemic species in Cape Verde, which have clear ecogeographic separation[21,22,30] and often retain inter-compatibility[21], so that the CVI *Arabidopsis* lineages could serve as a useful model for island endemic species more generally.

Although the Moroccan High Atlas population is genetically most similar to CVI across the genome (61%), there are prominent examples where it is not—including the chloroplast and the S-locus (Supplementary Figs. 4–6, Supplementary Note 1)—suggesting that an unsampled 'ghost' population best represents the outgroup. To obtain an upper (i.e., more ancient) bound on colonization time, we modelled the split between CVI and this 'ghost' population. We used multiple complementary approaches, including inference based on the joint site frequency spectrum, reconstruction of coalescence events across the genome, and comparisons to forward simulations[31–34]. These analyses revealed an initial separation between the Moroccan population and the CVI progenitor 'ghost' population at 40–60 kya, followed by colonization of CVI from the 'ghost' population as early as 7–10 kya (Supplementary Fig. 7, Supplementary Table 2, Supplementary Note 1).

To obtain a lower (i.e., more recent) bound on colonization time, we next examined coalescence time within CVI. Historical reconstruction[32,35] indicated that both islands were colonized through strong bottlenecks, which eliminated nearly all pre-existing variation (Fig. 2a, b). Using haplotype coalescence events we estimated the number of colonizers[34] and confidence intervals around these[36]. The estimated number of founders was 40 individuals (95% CI: 19–54) in Santo Antão and 48 individuals (95% CI: 30–66) in Fogo[34] (Fig. 2c). After the initial colonization, random effects of allele sampling (i.e., genetic drift) would have resulted in further reduction in diversity and sharing with ancestral populations. To quantify this effect, we ran simulations based on the inferred effective population sizes over time starting with 40 founders. These revealed that in the present-day population only 1.7 (95% CI: 0.6–3) variants in 10,000 are expected to have come from the original founding population. This implies that nearly all variation segregating in CVI results from mutations that occurred de novo after colonization.

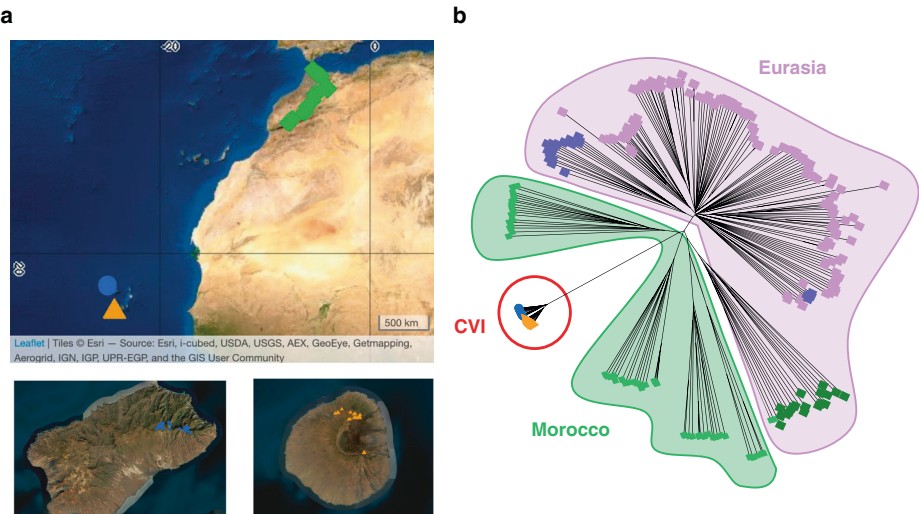

**Fig. 1 Population structure of Cape Verde *Arabidopsis*. a** Sample locations of sequenced lines from Morocco (green, $n = 64$) and the Cape Verde Islands, Santo Antão (blue, $n = 189$) and Fogo (orange, $n = 146$). The base map used derives from the World Imagery Map[117] and was accessed through the R package leaflet[118]. Values on the *x*-axis and *y*-axis represent, respectively, longitude and latitude. **b** Neighbour-joining tree showing relationship of CVI ($n = 336$, in blue and orange) to worldwide samples. Morocco: $n = 64$, in green; Eurasia: $n = 180$, including 20 samples, each from 9 representative clusters[26]. Divergent Iberian lines, or relicts, are shown in dark green, other Iberian lines (non-relicts) are shown in purple, and all other Eurasian lines in magenta. Source data are provided as a Source Data file.

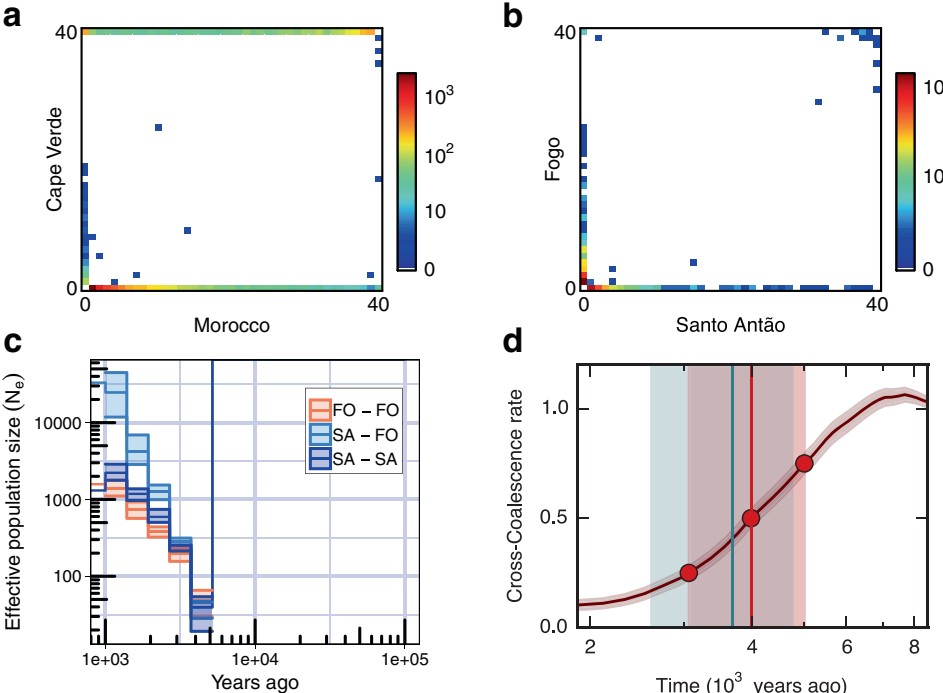

**Fig. 2 Demographic history of Cape Verde *Arabidopsis*. a** Joint site frequency spectrum between Cape Verde and Morocco and, (**b**), between the two islands, Santo Antão and Fogo. The colour scale represents the number of variants in each frequency class. **c** Historical population size trajectories within (SA-SA, FO-FO) and between islands (SA-FO) inferred from RELATE. Shaded areas represent the 95% CI calculated from the genome-wide distribution of coalescence events. **d** Estimated split times between islands from MSMC-CCR (point estimate at the vertical red line (4.0 kya), 0.25–0.75 cross-coalescence rate quantiles shown as shaded red area (3.1–5.0 kya)), and dadi (point estimate at the vertical blue line (3.7 kya), 95% CI as shaded blue area (2.6–4.8 kya)). Source data are provided as a Source Data file.

Between the two islands, patterns of variation differ, with Santo Antão displaying a higher proportion of private variation at segregating sites and Fogo displaying a higher proportion of private fixed variants (Fig. 2b). Consistent with this, we found evidence for deep population structure and restricted gene flow in Santo Antão, based on haplotype divergence among subpopulations. The overall pattern suggests early population subdivision followed by later population expansion across the island, with $N_e$ increasing sharply in the past 3 ky (Fig. 2c). In Fogo, the more arid island, there is no evidence of early separation into

subpopulations. Rather, we find a clear signal that after an initial moderate expansion (from approx. 48 individuals to 400 individuals) the population remained panmictic and restricted in size for approx. 830–940 years after colonization (Fig. 2c, Supplementary Fig. 8, Supplementary Table 3). Overall, our inference supports a model in which Santo Antão was colonized first (approximately 5–7 kya), and Fogo was colonized from Santo Antão approximately 3–5 kya[31,32,34] (Fig. 2c, d, Supplementary Fig. 8, Supplementary Notes 1 and 2). Our inferences clearly place the initial colonization of CVI well before colonization by humans, which only occurred approx. 560 years ago, implying that colonization occurred by natural (non-human) dispersal, e.g., by wind-mediated transport. Figure 3 provides a schematic of the

history that combines results from the different population genetic analyses.

**Moroccan climatic niche and suitability of CVI landscape.** To infer the suitability of the CVI climate to the colonizers when they initially arrived, we modelled the climatic niche of Moroccan *A. thaliana* and predicted suitability in CVI based on this model. We used Maxent[37] to model the factors that limit the distribution of *Arabidopsis* in Morocco based on georeferenced collection locations (Fig. 4a, Supplementary Note 3) and the set of bioclimatic variables listed in Supplementary Data 2. The main contributors to the model were the length of the growing season (38.7%), isothermality (20.2%), minimum temperature in the coldest month (18.4%) and maximum temperature in the warmest month (14.5%); (model AUC: 0.938 (std dev = 0.088); Fig. 4b; Supplementary Data 3, Supplementary Table 4). We predicted suitability of the CVI environment by projecting this model onto the CVI landscape. This analysis identified no suitable regions for Moroccan *Arabidopsis* in CVI (Fig. 4c). This may be expected given that distributions of climate variables taken from CVI collection locations are often outside of the range of those at Moroccan collection locations (Supplementary Fig. 2, Supplementary Data 2). Therefore, we also used an approach to examine the multivariate environmental similarity surface. The regions with highest climatic similarity from this analysis (Fig. 4d) are those where *Arabidopsis* can be found in Santo Antão and Fogo (Fig. 1a, Supplementary Fig. 1a, b). Although there is the possibility that at the time of colonization the climates were somewhat more similar or that the Moroccan population extended into more extreme climatic zones, based on our results using present-day data, there are large differences in many aspects of climate in CVI relative to Morocco. The overall low suitability and similarity of the CVI environment compared to that of the Moroccan population are thus consistent with the idea that the initial

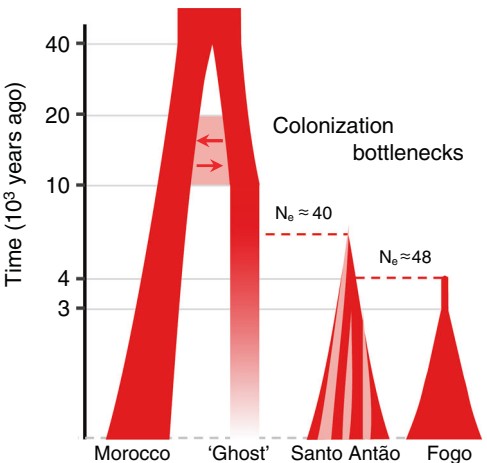

**Fig. 3 Schematic of the inferred history of CVI *Arabidopsis*.** $N_e$: effective population size; arrows denote migration. The y-axis is $\log_{10}$ transformed.

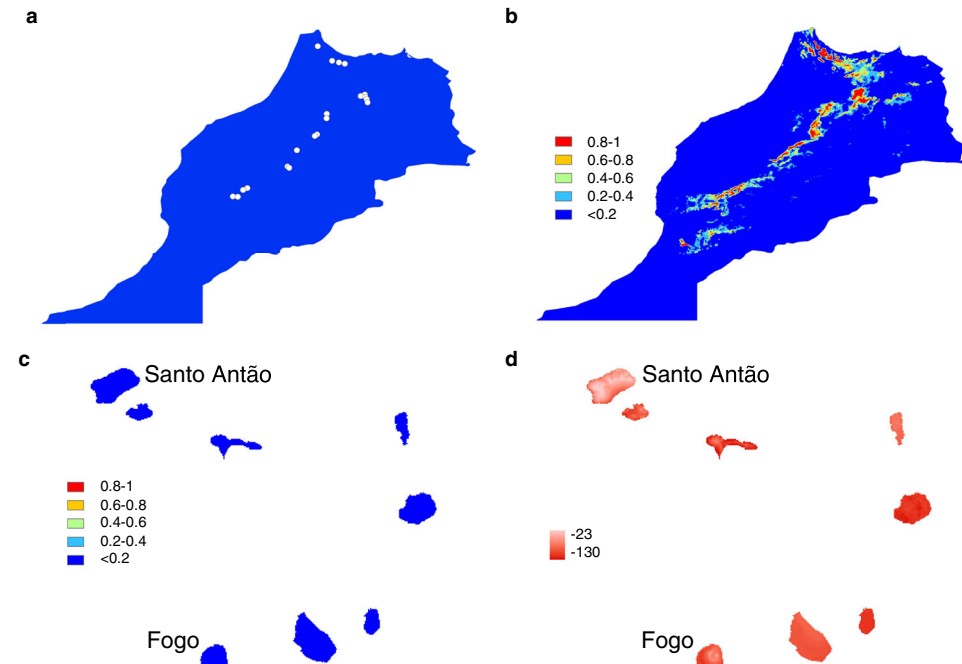

**Fig. 4 Moroccan and CVI predicted distributions. a** Moroccan *A. thaliana* occurrence locations and (**b**), predicted climate envelope within Morocco. In (**b**, **c**) colours represent the predicted probability of occurrence and habitat suitability, with blue indicating low probability and red high. **c** Predicted climatic suitability for Moroccan *Arabidopsis* in CVI and (**d**), predicted similarity of CVI climate to the Moroccan *A. thaliana* climate envelope expressed as a percentage of how dissimilar each point is in relation to the range of values used in the model. More negative (red) values indicate higher dissimilarity relative to Morocco. Maps were produced using ESRI ArcGIS.

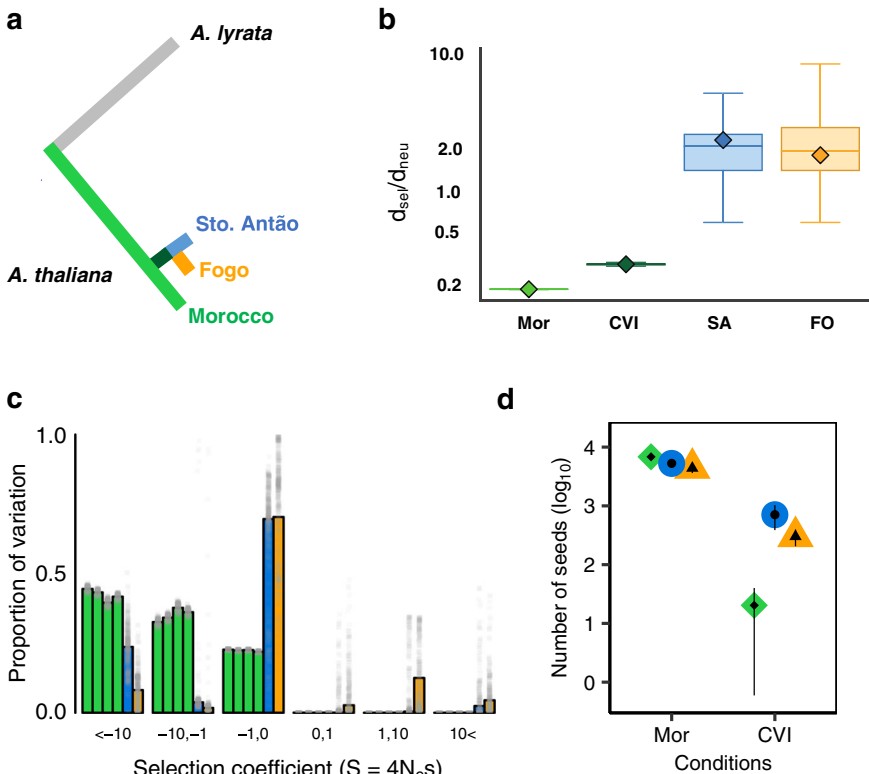

**Fig. 5 Population genetic signatures of adaptive evolution in CVI. a** Schematic of phylogeny separating branches examined in $d_{sel}/d_{neu}$ analysis. **b** Evolutionary rate ratios $d_{sel}/d_{neu}$ across populations (observed data shown as diamonds) with $n = 500$ bootstrap resampling replicates showing median (line), 1st and 3rd quartiles (bounds of box), minimum and maximum (whiskers). Mor: Morocco; CVI: branch between Morocco and CVI; SA: branch from the island split to Santo Antão; FO: branch from the island split to Fogo. **c** Distribution of fitness effects for Morocco (green), Santo Antão (blue), and Fogo (orange). Grey dots represent estimates from 500 bootstrap analyses. **d** Fitness scaled to seed number in CVI and Moroccan lines under CVI and Moroccan conditions. Medians per population are shown by the dots, and 95% CI by the whiskers. Y-axis values are $\log_{10}$ transformed. Source data are provided as a Source Data file.

colonizers would have been challenged by multiple aspects of the novel CVI environment.

**Evidence for adaptation based on functional genetic divergence and differential fitness.** Both drift and positive selection can contribute to genetic divergence. We used two approaches to investigate the role of adaptive evolution in CVI. The first is based on patterns of polymorphism and divergence within and between lineages and the second on an experimental test of relative reproductive success under CVI versus Moroccan conditions.

First, we examined evidence for positive selection on the branches of the phylogeny leading to the islands based on the relative fixation rate for mutations at amino acid replacement compared to synonymous substitutions. Specifically, we compared the ratio of nucleotide divergence at 0-fold nonsynonymous (putatively selected) to 4-fold synonymous (putatively neutral) sites, scaled to the number of sites at risk for each mutation (which we refer to as $d_{sel}/d_{neu}$, following[38]). This statistic is analogous to d$N$/d$S$[39] but excludes two- and three-fold degenerate sites, which are problematic to infer due to asymmetries in substitution rates. A value of unity is attained for $d_{sel}/d_{neu}$ when observed and expected substitution rates are equal, i.e., under the complete absence of selection (positive or purifying). Values less than unity imply purifying selection, and values greater than unity represent evidence for positive selection. We calculated whole-genome $d_{sel}/d_{neu}$ on the branch between Morocco and the most recent common ancestor of the two islands (i.e., variation fixed derived in CVI and absent from

Morocco) as well as on the branches leading to each individual island (i.e., variation private to a single island, and fixed there) (Fig. 5a, Supplementary Note 4). For comparison, we also calculated $d_{sel}/d_{neu}$ on the branch leading to the Moroccan *A. thaliana* population, which represents the core of the *A. thaliana* species[25], from the *A. lyrata* outgroup. We note that it was previously shown that pairwise $d_{sel}/d_{neu}$ comparisons between populations within a species (i.e., those that segregate for variation at an appreciable portion of the genome) are problematic[40]. However, given the phylogenetic separation between CVI populations and the Moroccan outgroup this is not relevant here. We found $d_{sel}/d_{neu}$ was greater than unity in both islands (Santo Antão: $d_{sel}/d_{neu} = 2.2$, Fogo: $d_{sel}/d_{neu} = 1.7$), consistent with strong positive selection on the nascent lineages, likely acting in concert with relaxed purifying selection (Fig. 5b). In contrast, on the Moroccan branch and on the branch of shared fixed divergence $d_{sel}/d_{neu}$ was significantly lower (Morocco: $d_{sel}/d_{neu} = 0.18$; MWW test, $W = 5 \times 10^5$, $p$-value $< 2.2 \times 10^{-16}$, Divergence branch: $d_{sel}/d_{neu} = 0.28$; MWW test, $W = 5 \times 10^5$, $p$-value $< 2.2 \times 10^{-16}$).

We further inferred the distribution of fitness effects (DFE)[41,42] based on segregating variation, or more specifically, the discretised distribution of scaled selection coefficients ($S = 4N_e s$, where $N_e$ is the effective population size and $s$ the selection coefficient). The DFE contained large peaks corresponding to nearly neutral effects ($-1 < S < 0$) and smaller peaks corresponding to strongly positive ($1 < S < 10$) and negative effects ($S < -10$) (Fig. 5c, Supplementary Note 4). In Fogo, fixed nonsynonymous mutations were prominent in the DFE, representing a classic

signature of positive selection at the clade level, while in Santo Antão, nonsynonymous mutations at intermediate to high frequency were more prominent, consistent with population stratification and/or local adaptation[43]. It should be noted that population history can impact estimates of $d_{sel}/d_{neu}$ so that these may be somewhat inflated due to possible fixation of deleterious variants under rapid population expansion[44,45]. Conversely, in Morocco, $d_{sel}/d_{neu}$ may be underestimated due to recent population bottlenecks[44]. It should also be noted that linkage disequilibrium and demography can violate assumptions of the DFE inference[42]. However, the method used here takes these effects into account using nuisance parameters, and we find a rather rapid LD decay in each island (Supplementary Fig. 3). While the limited numbers of fixed and segregating sites in the relatively young CVI lineages necessarily leads to large confidence intervals on our estimates (Fig. 5b, c), overall, the results are consistent with strong positive selection after a shift to a new adaptive optimum in the nascent CVI lineages.

Although population genetic approaches can provide evidence for positive selection, they make several assumptions. Therefore, we also tested for evidence of local adaptation in CVI and Moroccan clades based on evidence for higher relative fitness in local versus foreign environments. We propagated CVI and Moroccan lines in growth chambers set to match CVI and Moroccan environments (Supplementary Fig. 9a, b) and scored fitness (number of seeds produced). These experiments aimed to examine the fitness effects of climatic factors that differentiate CVI and Morocco and would not capture biotic or edaphic factors important for fitness. We tested for population, environment and population by environment effects using negative binomial GLM to correct for overdispersion. In the CVI environment, we found CVI lines performed significantly better than Moroccan lines ($\beta_{population}$ = 2.90, p-value = $3.58 \times 10^{-4}$). In the Moroccan environment, all lines performed better compared to the CVI environment, ($\beta_{pop-CVI}$ = 2.63, p-value = 0.0151; $\beta_{pop-Mor}$ = 5.86, p-value < $2 \times 10^{-16}$). There was no significant difference in fitness for the Moroccan and CVI lines in the Moroccan-simulated environment ($b$ = 0.337, p-value = 0.679). (Fig. 5d, Supplementary Data 4). Taken together these results highlight the challenging climatic conditions plants would have faced upon colonization of CVI, consistent with the results from the climate niche analysis (Fig. 4).

**Evidence for ongoing multivariate adaptation in Santo Antão.** Next, we examined the nature of adaptation in Cape Verde by capitalizing on over twenty years of studies on Cvi-0. We identified QTL, candidate genes and specific functional variants from a meta-analysis of 129 QTL mapping studies and associated fine-mapping studies conducted in a recombinant population produced from a cross between Cvi-0 and Ler-0[46] (Fig. 6a, Supplementary Data 5). These data set allowed us to ask whether genetic polymorphisms that underlie the observed trait divergence between Cvi-0 and other worldwide lines (with Ler-0 as the European representative) were present in the colonizing population or whether they represent variation that arose from de novo mutations after colonization. Based on the deep divergence between the RIL parents (Cvi-0 and Ler-0), we expected that most or all of the variants would be found on the long divergence branch that separates the two Cape Verde islands from continental populations. This expectation can be quantified based on the background level of variation: genome-wide, 99.23% of the variants that segregate between Cvi-0 and Ler-0 are fixed in CVI and therefore may have been present in the colonizing population. The remaining 0.77% are private to Santo Antão (the island of origin of Cvi-0; Supplementary Fig. 1) and absent in Fogo, and

therefore can be inferred to have originated in CVI as new mutations (Fig. 6b). The null expectation was that only a small proportion of functional variation (roughly equal to the genome-wide level) would be private to Santo Antão.

At QTL mapping intervals, which cover most of the genome, we found very slight and non-significant enrichment of private variation relative to the genome-wide proportion (1.02-fold enrichment, Poisson test p-value = 0.2723; Fig. 6c). This increased at candidate genes (1.30-fold enrichment, Poisson test p-value = 0.078) and became strongly significant at validated functional variants (87-fold enrichment, Poisson test p-value = $1.417 \times 10^{-10}$). Functional variants private to Santo Antão affect core genes involved in flowering and light signalling (CRY2 V367M[47], FRI K232X[48], GI L718F[49,50]), immunity against bacterial pathogens (FLS2 N452fs[51]), stomatal aperture and water use efficiency (MPK12 G53R[52]), chloroplast size (FtsZ2-2 G441fs[53]), and fructose sensitivity similar to ABA- and ethylene-signalling mutant phenotypes (ANAC089 S224fs[54]). These variants all segregate within Santo Antão at intermediate to high frequencies (between 0.43 and 0.89) and most are involved in functions that could underlie adaptation to the more drought-prone environment plants colonizing CVI would face. This suggests that adaptation on these variants is ongoing in Santo Antão. The strong enrichment of functional variation private to and segregating within Santo Antão implies that CVI *Arabidopsis* is adapting using variation that arose after colonization rather than variation inherited from North African ancestors. Further, the absence of these variants in *Arabidopsis* populations in Fogo implies that different genetic variants are involved in adaptation there.

To assess the effects of these seven private functional variants on fitness, we conducted a linear regression with these as predictors of fitness. All together they explain 22.58% of the within-island variation in fitness, which was significantly more than expected based on randomly sampled sets of seven variants across an LD-pruned genome (empirical p-value = $4.99 \times 10^{-4}$). Then, we used stepwise regression to identify the variants with the strongest effects on fitness. The best model based on the RMSE over 1000 bootstrap replicates explained 22.04% of the within-island variation and included two variants in flowering time pathway genes with significant effects, FRI K232X and *GI* L718F (Supplementary Table 5). Cvi-0 is known for its fast flowering time relative to many other populations[46,55]. Based on this, we focused specifically on the flowering time trait.

**Mapping and historical reconstruction reveal convergent genetic adaptation to reduce flowering time.** We scored flowering time as days to bolting in plants grown in simulated CVI conditions. We found that plants from both islands flowered significantly earlier than Moroccans (MWW test, $W$ = 1620, p-value < $2.2 \times 10^{-16}$; Fig. 7a) and the majority of Moroccan lines never bolted in CVI conditions, resulting in a strong negative association between flowering time and fitness (Spearman's rho = $-0.85$, p-value < $2.2 \times 10^{-16}$; Fig. 7b). This is consistent with previous suggestions that reducing flowering time may allow escape from drought and provide an important fitness advantage[56–58]. To ask whether early flowering in the two islands results from the same or different variants, we examined segregation in three inter-island F2 populations (Fig. 7c, Supplementary Note 5). In each of these, flowering time was transgressive with some individuals flowering as early or earlier than the parents and some flowering much later (two-tailed Dunnett's tests with Fisher's method, $S$ = 67.187, p-value = $1.54 \times 10^{-12}$). Taken together, these results imply that flowering time was reduced in CVI by convergent evolution involving mutations at different loci in the two islands.

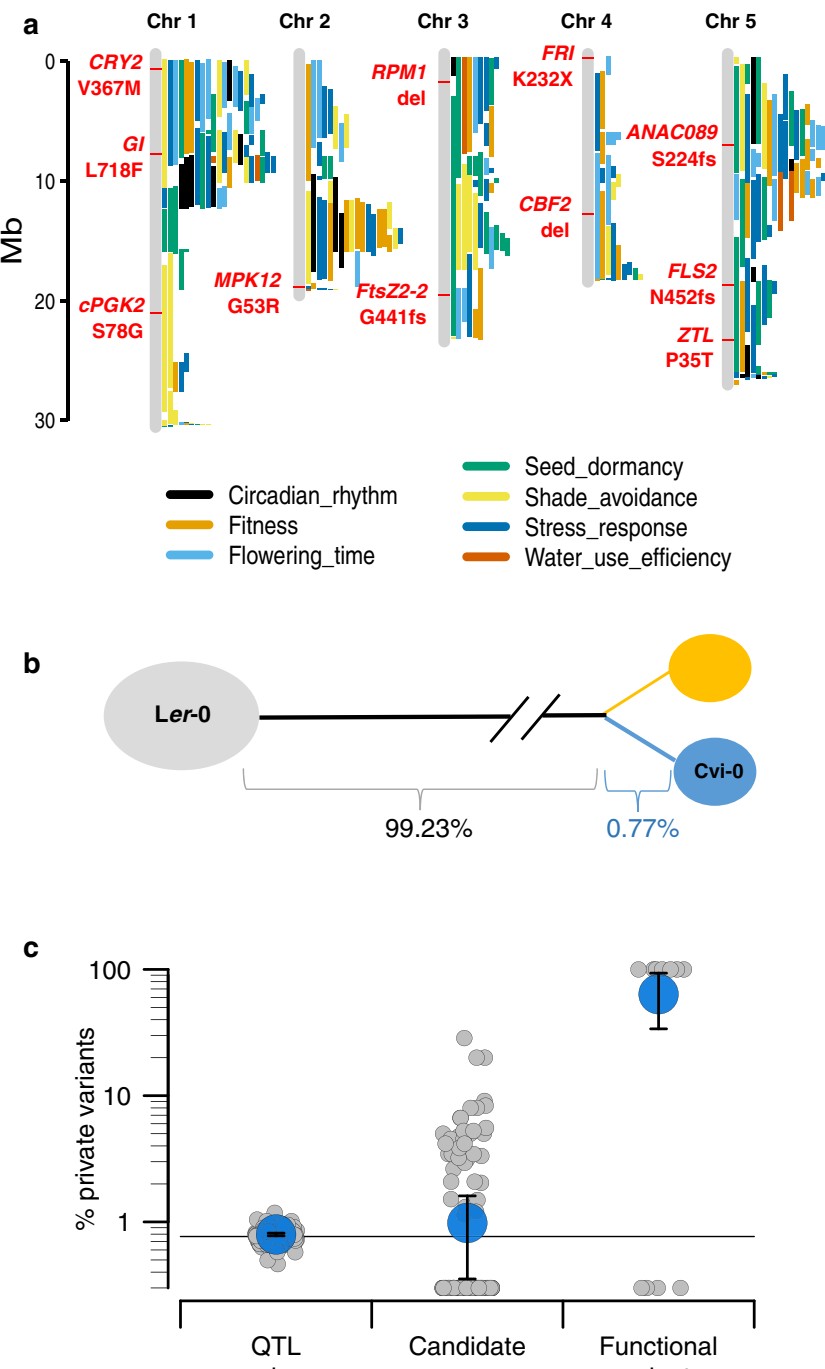

**Fig. 6 Evidence of multivariate selection from 129 QTL mapping analyses using Cvi-0. a** Representative subset of previously identified QTL in Cvi-0 x Ler-0 RILs, with validated functional variants in red. Each segment along the chromosomes represents one QTL region with colours representing phenotypic classes. **b** Schematic of the relationship between the RILs parents, with branch lengths proportional to the percentage of genome-wide variation fixed in CVI and segregating in Santo Antão. **c** Percentage of variation private to Santo Antão in QTL regions (n = 121), candidate genes (n = 142) and functional variants (n = 11). Horizontal line shows the genome-wide average. Each grey dot represents a single QTL, candidate gene or functional variant, blue dots represent the average, and error bars represent the 95% CI. Source data are provided as a Source Data file.

To identify the loci responsible for reduced flowering time, we performed GWAS using a linear mixed model (LMM) to account for population structure[59] (Supplementary Note 5). In the Santo Antão population, we identified a single peak containing a nonsense variant, K232X, in FRIGIDA (FRI, AT4G00650), which results in faster flowering through loss of the vernalization (cold) requirement[48] (Fig. 7d). This variant explained 46.4% of the genetic variance in flowering time and 11.4% of the heritable

variance in fitness. In the natural population, FRI 232X was associated with a 34-day decrease in flowering time (MWW test, $W = 7$, p-value $< 2.2 \times 10^{-16}$), and a 140-fold increase in seed number (+387 seeds; MWW test, $W = 4541$, p-value $= 7.18 \times 10^{-14}$; Fig. 7e). To further test whether loss of FRI was likely responsible for this effect, we compared a Col-0 transgenic line with a functional FRI allele to that with a non-functional FRI allele in the same environment and measured flowering time.

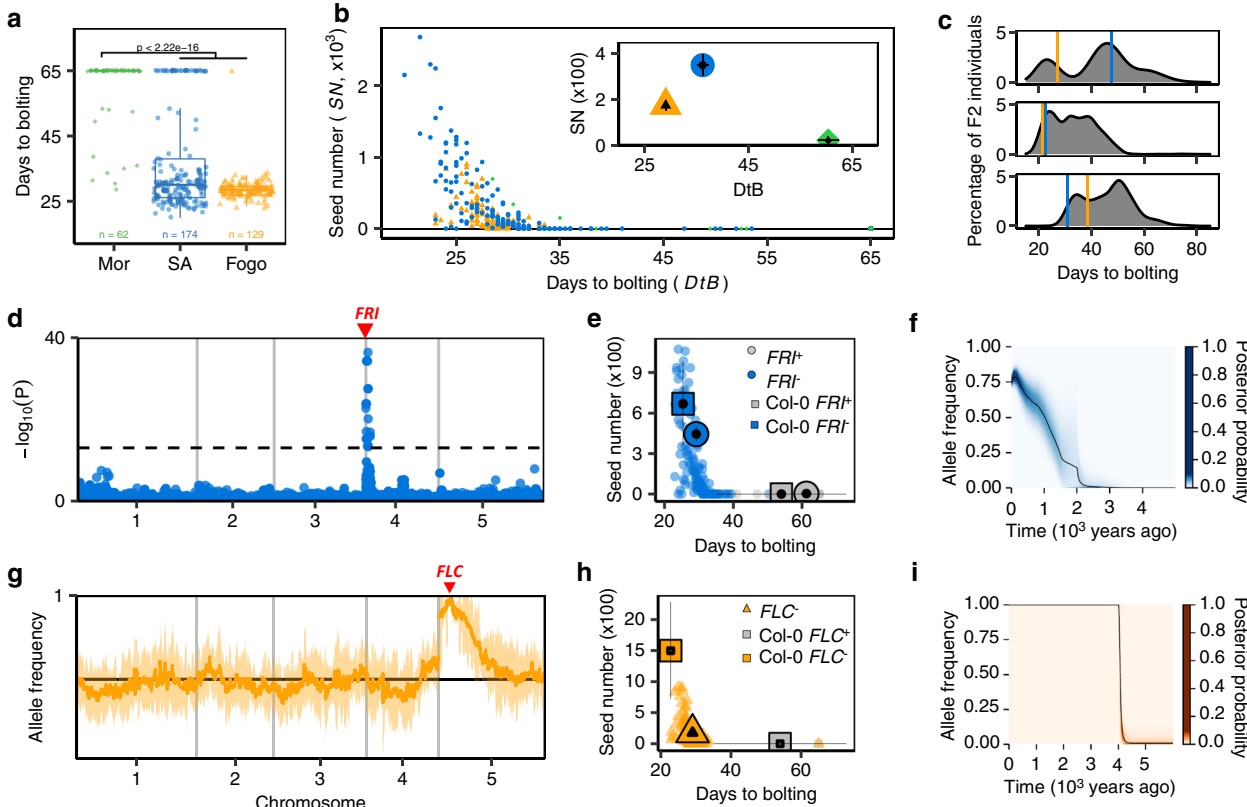

**Fig. 7 Adaptation through parallel reduction in flowering time in CVI. a** Bolting time in CVI relative to Morocco with two-sided MWW test to compare distributions ($p$-value $< 2.2 \times 10^{-16}$). Boxplots show median (centre), 1st and 3rd quartiles (lower and upper bound, respectively). Whiskers represent 95% CI. **b** Days to bolting versus fitness (seed number); inset: means and 95% CIs. **c** Bolting time is transgressive across islands in three inter-island F2 populations. Vertical lines represent medians of days to bolting across replicates of the Santo Antão (blue) and Fogo (orange) parents. **d** Bolting time GWAS in Santo Antão. Dashed line represents Bonferroni-corrected genome-wide significance. **e** Effects of *FRI* alleles on bolting time and seed number under simulated CVI conditions ($n = 189$). Small symbols represent individual lines, large symbols population means with 95% CI (whiskers). **f** Inferred allele frequency trajectory of the derived FRI 232X in Santo Antão. The black curve represents the posterior mean of the allele frequency and the coloured area the posterior distribution. **g** Frequency distribution of the Fogo allele in an early flowering bulk from an inter-island F2 population reveals a peak at *FLC*. Median frequency per window (line) and one standard deviation (shading) are shown. **h** Effects of *FLC* alleles on bolting time and seed number under simulated CVI conditions ($n = 146$) as in (**e**). **i** Inferred allele frequency trajectory of the derived FLC 3X allele in Fogo, as in (**f**). Source data are provided as a Source Data file.

We found that the effect is similar to that of the Santo Antão FRI 232X variant, (flowering time: $-27$ days, fitness: $+669$ seeds; MWW test $W = 0$, $p$-value $= 3.85 \times 10^{-3}$; $W = 37.5$, $p$-value $= 8.86 \times 10^{-3}$, respectively; Fig. 7e), further supporting the role of FRI 232X in flowering time reduction. FRI 232X is present at high frequency across all populations in Santo Antão except the early-diverging Cova de Paúl population, where it is completely absent (Supplementary Fig. 10). Coalescent reconstruction[34] of the history of FRI 232X indicated that the allele arose between 2.14 kya (95% CI: 1.62–2.72 kya) and 2.9 kya (95% CI: 2.14–3.74 kya) and rapidly spread across the island, with fixation likely restricted by barriers to gene flow (Supplementary Fig. 10). Based on the inferred frequency trajectory, we estimated that selection was maximized at 2–4 kya with a selection coefficient of $s = 4.56\%$ (Supplementary Table 6). The timing of the spread of FRI 232X is roughly coincident with the inferred expansion of *Arabidopsis* into the drier Espongeiro region of the island[34,60] (Fig. 7f).

In Fogo, the more arid island, all individuals flowered early with low variance (mean time to flowering $= 29.05$ days, SD $= 5.33$ days). This suggested that at least one genetic variant underlying reduced flowering time was fixed in Fogo. Trait segregation in an inter-island

F2 population (where FRI 232X was absent) exhibited a bimodal distribution with a 1:3 ratio (Fig. 7c top) and there were no major peaks in GWAS (Supplementary Fig. 11, Supplementary Note 5), indicating the presence of a single large effect early flowering allele. Sequencing the bulk of early flowering F2 individuals revealed a single region where the frequency of Fogo alleles reached 100%, corresponding to FLOWERING LOCUS C (*FLC*, AT5G10140; Fig. 7g). FLC is a central floral repressor that regulates genes responsible for the transition from the vegetative to the reproductive state and is regulated by FRI[61]. We identified a premature truncation mutation in FLC (R3X), which is fixed in Fogo and absent from Santo Antão, and confirmed by qRT-PCR and genetic complementation that this mutation causes loss of function (Supplementary Fig. 12, Supplementary Note 5). This variant decreased flowering time by 27 days (based on the difference in modes in the F2 population, MWW test, $W = 0$, $p$-value $< 2.2 \times 10^{-16}$), comparable to Col-0 $FRI^+FLC^-$ ($-31$ days; MWW test, $W = 25$, $p$-value $= 0.0107$; Fig. 7h). Similarly, loss of function in the Col-0 background (Col-0 $FRI^+FLC^-$) resulted in higher seed production relative to Col-0 $FRI^+FLC^+$ in simulated CVI conditions ($+1498$ seeds; MWW test, $W = 0$, $p$-value $= 7.5 \times 10^{-3}$). Coalescent reconstructions and inferred frequency trajectories of FLC 3X indicated that it arose soon

after colonization (between 3.31 kya (95% CI: 2.82–3.96 kya) and 4.72 kya (95% CI: 3.56–6.66 kya)) and was associated with strong positive selection[34,60] ($s = 9.27\%$; Fig. 7i, Supplementary Fig. 13, Supplementary Table 7, Supplementary Note 5).

In summary, loss of function mutations that greatly reduced flowering time appeared independently in Santo Antão (FRI 232X) and Fogo (FLC 3X) and their origins are temporally associated with initial increases in effective population size on the two islands (Fig. 2c). Because we take the inferred change in population size into account in our estimates of selection coefficients, these would be underestimated in the case that the variants themselves allow establishment and spread of populations across CVI. This may explain why the selection differentials estimated in simulated CVI environments for FRI and FLC loss of function variants are larger than the selection coefficients inferred from population genetic data. In Santo Antão, FRI 232X appears to have provided a strong selective advantage (Fig. 7e, f), likely enabling population expansion into drier regions of the island. In the more arid Fogo environment, the initial population appears to have been highly constrained in both size and breadth and there is a remarkable overlap in the estimate of the time when FLC 3X arose and fixed in Fogo and the initial increase in population size there (Supplementary Figs. 14, 15). The early appearance of these de novo variants is consistent with a role in evolutionary rescue of the nascent populations through reduced time to flowering.

**Extinction risk and adaptation via large effect mutations.** Colonization of a new environment brings with it multiple challenges. Colonization events are often associated with strong bottlenecks, reducing standing genetic variation available for adaptation. When combined with a sudden and severe change in the selection regime, as may often accompany long-range colonization, extinction risk is high[7,62,63]. This is because the expected waiting time for a beneficial mutation is likely to be greater than the expected time to extinction in a small maladapted colony[63]. Escape from extinction under this scenario is possible but relies on chance mutational events.

Theory predicts that when selection is strong and mutational input is low (i.e., a strong selection weak mutation (SSWM) regime), the first steps of adaptation are likely to occur through large effect mutations[8,23,64–68]. Conversely, when mutational input is high and selection is weak (i.e., a weak selection strong mutation (WSSM) regime), adaptation is likely to occur through more, smaller effect variants. Specifically, the SSWM model is expected to hold when (i) the total number of mutations that enter a population each generation is limited ($U_b \ll 1/4N_e$, where $N_e$ is the effective population size and $U_b$ is the genome-wide per-individual beneficial mutation rate for the focal trait) and (ii) selection is strong relative to drift ($s \gg 1/4N_e$).

We asked where the CVI case fits in relation to the SSWM and WSSM models. First, we approximated the genome-wide mutation rate for the adaptive phenotype: very early flowering through loss of vernalization. Then, we applied our inferences about historical population size and selection coefficients to examine the fit of adaptation in CVI to these models (details in Methods). We collated molecular information about the focal trait to produce a rough approximation of $U_b$ for coding and regulatory changes (Supplementary Note 6), resulting in an estimated $U_b = 1.54 \times 10^{-6}$ mutations per site per generation. Estimates of $s$ from reconstructed frequency trajectories were well above $1/4N_e$, and estimates of $U_b$ were well below $1/4N_e$ in both Fogo ($s = 0.093$ and $1/4N_e = 5.21 \times 10^{-3}$) and Santo Antão ($s = 0.046$, $1/4N_e$ ranging from $2.5 \times 10^{-4}$ to $5 \times 10^{-4}$; Supplementary Note 6), implying a SSWM regime. We also conducted forward simulations modelled after the Fogo

population that incorporated the stochastic effects of drift across a range of plausible selfing rates (90%–99%; Supplementary Fig. 14, Supplementary Table 8). Taken together, our results imply that the scenarios in CVI are predictable and consistent with the SSWM regime, where mutation is limited and adaptation and establishment after initial colonization relies on sweeps of large effect alleles[5,8,64,69].

## Discussion

We found several lines of evidence that adaptation was crucial for establishment of *A. thaliana* in CVI. First, early colonists from North Africa faced a severe climatic challenge (Fig. 4). Second, population genetic data revealed an increased rate of nonsynonymous substitution on the branches leading to the current island populations (Fig. 5b) as well as an excess of intermediate to high-frequency functional variants within Santo Antão (Figs. 5c, 6c). Third, we found evidence for higher relative fitness of Cape Verdean accessions compared to Moroccans in simulated conditions (Fig. 5d). The time to flowering was strongly associated with this fitness differential (Fig. 7b). Mapping (Fig. 7d, g) and evolutionary reconstructions (Fig. 7f, i) revealed that in each island, a variant that drastically reduced flowering time through loss of the vernalization (cold) requirement (FRI 232X, FLC 3X) was driven to high frequency by strong positive selection. Overall, the dynamics for both FRI and FLC mutations are consistent with a *strong selection, weak mutation* regime[64–66], where adaptation occurred by convergent loss of the vernalization requirement (Supplementary Note 6).

In Santo Antão, strong selection favoured early flowering (Fig. 7a, f) and was linked to establishment across the drier regions of the island. In more arid Fogo, population size increased in the same time frame when FLC 3X arose and fixed (Supplementary Fig. 15). Given the clear fitness advantage of reduced flowering time in CVI (Fig. 7d), this concordance strongly suggests that FLC 3X enabled escape from extinction in Fogo (Supplementary Fig. 14-15).

Functional variation in FRI and FLC is widespread in natural populations of *A. thaliana*[48,70–75] and in homologues across species[76–82]. Adaptive mechanisms have been suggested to explain the prevalence of nonsynonymous variation in *FRI*[83] and clinal patterns in flowering time in European *A. thaliana* populations[75,84,85]. Here, at the southern extreme of the *Arabidopsis* species distribution, the natural experiment in the isolated Cape Verde Islands allowed us to definitively connect mutations that occurred in parallel at FRI and FLC with adaptive divergence. Evolutionary convergence in this case highlights the importance of these two genes in adaptation to growing season length and aridity.

Our population genetic analyses (Fig. 5a, b) and investigation of patterns at known functional loci (Fig. 6c) further suggest that adaptation in Cape Verde was multivariate and involved many loci and traits. Some of these would be reflected in fitness differentials in the simulated CVI and Moroccan environments. But others—such as differences in biotic and edaphic factors—would not be captured in our simulated conditions. Future work in these *Arabidopsis* island lineages will be necessary to better characterize the multivariate history of adaptation here.

Detailing the mechanisms of adaptation after a sudden environmental shift provides useful information for forecasting and ameliorating risk for vulnerable populations and species. Small, isolated populations that confront abrupt environmental change face high extinction risk[7,11,62,63]. Adaptive escape from extinction in these cases is a race with the clock, in particular when standing variation is not available. Adaptation in CVI fits well with the theoretical concept of an adaptive walk[24,64–66,86,87], in which a

small, mutation-limited population faced a new environment far from its previous adaptive optimum and, due to the lack of standing variation, initially relied on beneficial mutations to adapt (Supplementary Note 6). This is in-line with models of rapid adaptation and evolutionary rescue from large effect mutations[6,24,67,86]. Our findings are reminiscent of work in laboratory-based microbial experiments showing that independent bouts of evolution often use the same paths[68,88–94]. Further, they suggest that adaptation to increasing aridity and shorter growing seasons—which are expected to be common under global climate change—is predictable. Therefore, our findings could also be relevant in efforts to tailor crops to drought-prone environments.

## Methods

**Plant material**. We collected plants over a series of field expeditions between 2012 and 2019 on Santo Antão and Fogo, the two islands where *A. thaliana* had been documented in herbarium records. In total, we present data for 335 lines from CVI (Supplementary Data 1, Fig. 1a, Supplementary Method 1), including 189 lines from 26 stands across four regions in Santo Antão (Cova de Paúl, Lombo de Figueira, Pico da Cruz and Espongeiro), and 146 lines from 18 stands across three regions in Fogo (Lava, Monte Velha and Inferno). The 62 Moroccan lines used in the study were first presented in[95] and were sequenced in[25].

**Climate data**. Climate data used in our analyses were retrieved from the World-clim Project[96] and CGIAR Consortium (CGIAR-CSI)[97] (Supplementary Method 2).

**Sequencing**. We sequenced the 335 Cape Verde Islands lines and Cvi-0 using Illumina Hi-Seq and HiSeq3000 machines (Supplementary Method 3). Genomic DNA was extracted using the DNeasy Plant Mini kits (Qiagen), fragmented using sonication (Covaris S2), and libraries were prepared with Illumina TruSeq DNA sample prep kits (Illumina), NEBNext Ultra II FS DNA Library Prep Kit (New England Biolabs) and NEBNext Ultra II DNA Library Prep Kit (New England Biolabs). Libraries were immobilized and processed onto a flow cell with cBot (Illumina) and subsequently sequenced with 2x 100–150 bp paired end reads. We assessed DNA quality and quantity via capillary electrophoresis (TapeStation, Agilent Technologies) and fluorometry (Qubit and Nanodrop, Thermo Fisher Scientific). Due to changes in product availability over time, there were some slight differences among sequencing runs.

**SNP identification and genotyping**. We aligned the raw Illumina sequence data for the CVI samples together with previously sequenced Eurasian[98] and Moroccan samples[25] to the *Arabidopsis* TAIR10 reference genome and we identified and genotyped variants (Supplementary Method 4, https://github.com/HancockLab). To eliminate false variant calls due to duplications not represented in the reference genome, we filtered out genomic regions with coverage higher than twice the genomic average. Further, for trait mapping, we used a pipeline based on GATK[9] for the additional analyses of short indels using a modified version of the best practices workflows for germline short variant discovery (https://github.com/HancockLab/SNP_and_Indel_calling_Arabidopsis_GATK4). Average coverage across samples was 19.4x (range from 9.3x to 51.7x) after alignment to the TAIR10 reference genome.

**Plant growth and phenotyping**. For all experiments, seeds were stratified in the dark in Petri dishes on water-soaked filter paper for one week at 4 °C prior to sowing. After stratification, seeds were sown in 7 × 7cm pots containing a standard potting compost mix. Four seeds were sown per pot and plants were thinned to one plant per pot, after germination. Further details can be found in Supplementary Method 5.

We simulated the CVI growing season in a custom Bronson growth chamber based on hourly environmental data at a collection site (Supplementary Fig. 9), where we measured air and soil temperature, air humidity and precipitation using data loggers. The experiment began with September 1, 2016 conditions, when we observed plants germinating at the field site. Photoperiod was set to track daylength (number of sunlight hours) in CVI. We simulated dawn and dusk by increasing light intensity by 50 µM every 15 minutes until 200 µM (full light) and decreasing it by 50 µM every 15 min until dark, respectively. At the same time points, far-red light decreased from 50 to 0 µM at dawn and increased from 0 to 50 µM at dusk. Based on precipitation data from the field, we withheld water starting 26 days after sowing. To mimic the gradual decrease in soil moisture levels we observed in the field, we used capillary mats to buffer the drought. Moroccan conditions were simulated based on matching to temperature and photoperiod in relevant locations within the Moroccan Atlas mountains[95] (https://www.worldweatheronline.com/morocco-weather.aspx). For this condition, photoperiod was set to 12 h and plants

were submitted to an eight-week cold period (4 °C) starting two weeks after sowing, to match winter temperatures.

In CVI simulated conditions, we propagated 174 Santo Antão and 129 Fogo lines in four replicates each, and 64 Moroccan lines in two replicates each. Based on results from a preliminary pilot experiment, two mutants were included: Col-0 with a functional *FRI* introgressed from the Sf-2 line (Col-0 *FRI*-Sf2, shown as Col-0 *FRI⁺FLC⁺*)[61], and Col-0 *FRI*-Sf2 with a non-functional *FLC* allele (Col-0 *FRI*-Sf2 *flc*-3, shown as Col-0 *FRI⁺FLC⁻*)[61] as well as Col-0 as a control. The plants were organized in a randomized block design and Aracon tubes were added when the plants flowered to allow for the total set of seeds to be collected. We scored flowering time, bolting time, time to anthesis, number of days until the stem reached 3 cm, and the number of rosette leaves at bolting, as in[99] as well as fitness. For downstream analyses, bolting time was used as a proxy for flowering time. The experiment was terminated ten weeks after sowing, when plants no longer produced new flowers or seeds. Plants that had not bolted at the end of the experiment were conservatively scored as bolting at 65 days (following[95]). A total number of seeds per individual was scored as a measure of fitness. Seeds were counted using the Germinator plugin[100] implemented in ImageJ v.1.40[101]. In Moroccan-simulated conditions, we propagated the 64 Moroccan lines in four replicates together with a set of eight representative Cape Verdean lines (four from Santo Antão and four from Fogo) in eight replicates each. To assess fitness differences between populations under CVI and Moroccan-simulated conditions, we collected the complete sets of seeds produced per individual. In the CVI simulated conditions, where total seed numbers were limited, we counted the seeds, and from the Moroccan conditions we weighed seeds and estimated the counts based on the weight of 100 seeds.

**Population structure, diversity, and demographic reconstruction**. We evenly subsampled the 13 genetic clusters identified previously on the continents (nine in Eurasia[10], four in Africa[8]) and the two Cape Verdean Islands populations to 20 samples per cluster to avoid biases due to differences in sample size across populations. The only exceptions were the Moroccan Rif, North Middle Atlas and High Atlas populations where fewer samples are available (respectively, 8, 13, and 16). We pruned the data set for short-range linkage disequilibrium <--indep-pairwise 50 10 0.1> and for missing data <--geno 0> using PLINK v.1.90 and removed multi-allelic variants. We produced neighbour-joining trees using the R package *ape* v.3.5[102] (https://github.com/HancockLab/CVI).

We used custom scripts to estimate nucleotide diversity ($\theta$) in CVI, Morocco and Eurasia by computing Tajima's ($\theta_\pi$) and Watterson's estimators ($\theta_w$), as well as for deriving the site frequency spectra (SFS) (https://github.com/HancockLab/CVI). The joint site frequency spectrum (JSFS) between islands was computed on a subsampled set of 40 individuals per island. We excluded sites with more than 5% missing data, CpG sites, due to their hypermutable nature, pericentromeric regions, which are rich in satellite repeats, and other repeat regions identified with Heng Li's SNPable approach (http://bit.ly/snpable). The JSFS between CVI versus Morocco was computed using both CVI islands together and was polarized to the outgroup species *Arabidopsis lyrata*. We aligned short-read data for 27 *A. lyrata* genomes to the *A. thaliana* reference genome (TAIR10) and retained for analyses only SNPs that were not polymorphic in *A. lyrata* and for which there were no missing data. To polarize the JSFS between islands, we reconstructed the most likely ancestral state at every SNP based on variation in Morocco, the best modern representative of the original colonizing lineage. At sites that were fixed in Cape Verde, a state was assigned as ancestral if it was found in Morocco; otherwise, it was assigned as derived. We used the same approach for sites that were polymorphic in Cape Verde. In cases where both alleles were found in Morocco, a missing value was assigned for the ancestral state.

Linkage disequilibrium (LD) was assessed in PLINK[103,104] by computing the correlation ($r^2$) in frequency across pairs of SNPs up to a distance of 10 kb. SNP pairs were clustered into bins of 1 kb and $r^2$ values within each bin were averaged (Supplementary Method 6).

We inferred haplotypes across the genome, separated by historical recombination events, and screened a set of potential donor populations for the closest relative at each haplotype using Chromopainter v.0.0.4[105]. We used a representative subset of 148 CVI genomes from the two islands. As donors, we used the 13 mainland clusters previously identified (nine in Eurasia[26], four in North Africa[25]). Each donor population was randomly subsampled to 20 samples 100 times, and for each subsampling we ran Chromopainter ten times for a total of 1000 replicated analyses of each Cape Verdean genome (https://github.com/HancockLab/CVI).

We inferred colonization time by obtaining an upper bound based on the minimum coalescence time between CVI and Morocco, and a lower bound based on the maximum coalescence time within the CVI clade (Supplementary Methods 7 and 8).

We inferred split times between the two Cape Verde Islands, among subpopulations within islands and between CVI and Morocco using the cross-coalescence rate (CCR) statistic in the MSMC2 framework[17,18] as well as with dadi v.2.1.0[32], which derives estimates for parameters based on fitting the JSFS. For both methods, we assumed a generation time of one year and a mutation rate of 7.1 × 10⁻⁹ [106]. MSMC2-CCR consists of comparing the rate of inferred coalescences between groups to the average rate within groups across time. CCR decays from

one towards zero as populations split from each other. For analyses with MSMC2-CCR, we combined the effectively haploid genomes to produce artificial diploids. Diploids were created by combining lines from the same stand to avoid biases due to structure. We used the eight-haplotype implementation of MSMC2, which has the best resolution for recent events (up to approx. 1 kya in our system). For the inference of split parameters in dadi v.2.1.0[32], we used intergenic JSFS, which are less likely to evolve under strong selection. We estimated parameters between the two Cape Verde islands and between CVI and Morocco using four demographic models. For each model and population pair, we conducted the analysis 1000 times with up to 50 iterations to infer confidence intervals.

We used three complementary approaches to model the demographic history within the archipelago including the timing of colonization and severity of the associated bottlenecks. First, we ran RELATE[34] and COLATE[36] under a haploid model using the module 'EstimatePopulationSize' to reconstruct $N_e$ over time based on inferred coalescence events within each island population. In addition, we fit a model to the data using forward-in-time, individual-based simulations from Slim3[21]. We also conducted inference based on phylogenetic analysis of the non-recombining chloroplast locus to check for agreement at this locus.

**Niche modelling**. We performed niche modeling in Maxent[37] based on the bioclimatic variables described in Supplementary Table 1. We used standard default parameters with jackknife resampling to estimate the importance of each variable on the model. We built a model to predict the suitability across the Cape Verde archipelago for colonization by *A. thaliana* from the Moroccan range, and to identify the regions within Cape Verde that are most similar to the Moroccan habitat (Supplementary Method 9).

**Testing for evidence of adaptive evolution**. We used custom scripts (https://github.com/HancockLab/CVI) to compute the $d_{sel}/d_{neu}$ ratio, defined as the rate ratio of 0-fold nonsynonymous to 4-fold synonymous substitutions, scaled by the number of sites at risk for each category. Genome-wide, after discounting sites with more than 5% missing data, the number of sites at risk for 0-fold and 4-fold mutations were respectively 5967270 and 1332660. To address the divergence branch between the two islands and the mainland, we used mutations that are fixed derived in Cape Verde and absent from Morocco. To address the branches leading to each individual island, we used mutations that are fixed derived in one island and absent from the other island and Morocco. We used the spectra at zero- and four-fold degenerate sites to infer the distribution of fitness effects (DFE) with polyDfe v.2.0[41] using default parameters <-m C -o bfgs>. We ran the analysis independently for the two CVI islands (11 samples in Fogo and 13 in Santo Antão), and Morocco. For both analyses, confidence intervals were estimated based on resampling. Further details can be found in Supplementary Method 10.

**Identifying QTLs, candidate genes, and functional variants**. We conducted a literature review of studies that used the Cvi-0 x L*er*-0 RILs and, based on these studies together with fine-mapping and downstream functional analyses, we compiled lists of candidate genes and validated functional variants (Supplementary Method 11).

**Trait mapping**. We conducted genome-wide association analysis (GWAS) using a univariate linear mixed model while accounting for population structure with a mean-centred kinship matrix <-gk 1> using the flag <-lmm 4> in GEMMA[99]. Input files for this analysis were generated on GATK genotypes, which included indel calls, using VCFtools[107] and PLINK[104]. Mapping was conducted based on the median phenotype across replicates per genotype (https://github.com/HancockLab), since no block effect was detected across the chamber (Supplementary Method 12).

For bulked segregant analysis, we propagated an inter-island F2 population (S5-10 x F13-8, $n = 488$), in which the ancestral allele FRI K232 was fixed), under simulated CVI conditions. Because early flowering segregated at an approximately 1:3 ratio (indicating a single recessive locus), we sampled leaf tissue from the 25% early tail of the F2 ($n = 108$). We extracted DNA using a DNeasy Plant Mini kit (Qiagen), assessed DNA quality and quantity with Qubit and Nanodrop (Thermo Fisher Scientific), prepared a single library using NEBNext Ultra II FS DNA Library Prep Kit (New England Biolabs) and sequenced it to 50x coverage using the Illumina HiSeq3000 platform. We called variants against the TAIR10 reference assembly using a GATK pipeline[108] (https://github.com/HancockLab/CVI), retaining only biallelic variants. We identified window(s) where the median allele frequency dispersion was greater than 95% and annotated variants within candidate region(s) using SnpEff v.3.0[109]. These are listed in Supplementary Data 6.

**Functional validation**. We measured *FLC* expression in a representative set of eight Cape Verdean and six Moroccan lines as well as in the Col-0 reference line, a modified Col-0 with a functional *FRI* introgressed (Col-0 *FRI*-Sf2, shown as Col-0 *FRI*+*FLC*+), since FRI affects *FLC* mRNA levels[71,72], and Col-0 *FRI*-Sf2 with an *FLC* knock-out (Col-0 *FRI*-Sf2 *flc-3*, shown as Col-0 *FRI*+*FLC*−)[61]. We grew three replicates of each genotype under CVI simulated conditions (12 h light, 20 °C at day, 14 °C at night) and assessed mRNA levels by qRT-PCR on a LightCycler 480

instrument (Roche) using the $2^{-\Delta\Delta Ct}$ method (Applied Biosystems) and PP2A (AT1G13320) as a reference gene. Primers used in this experiment are listed in Supplementary Table 9 and further details in Supplementary Method 13.

We performed genetic complementation tests for *FLC* by crossing four individuals from Fogo (each with the FLC 3X allele) to Col-0 *FRI*-Sf2 plants with and without a functional *FLC* allele (Col-0 *FRI*-Sf2, referred to as Col-0 *FRI*+*FLC*+, and Col-0 *FRI*-Sf2 *flc-3*, referred to as Col-0 *FRI*+*FLC*−, respectively). We also crossed the mutants (Col-0 background) to obtain a heterozygous F1 at *FLC*. We grew four replicates of each parent and F1 per cross and scored bolting and flowering time in 12 h standard greenhouse conditions (Supplementary Method 14).

**Historical reconstruction of evolution of *FRI* and *FLC* loci**. We used RELATE v1.1.4 to infer the genealogical trees for the derived alleles FRI 232X (Chr4:269719) and FLC 3X (Chr5:3179333) and we used CLUES[60] to infer the frequency trajectory and selection coefficient for the derived FRI 232X and FLC 3X alleles (Supplementary Method 15). Selection coefficients were inferred relative to the reconstructed demographic history for each island (Supplementary Tables 10, 11). We calculated the fit to strong selection weak mutation (SSWM) and weak selection strong mutation (WSSM) models of evolution[64–66] using an estimate of the genome-wide mutational target size based on molecular studies[71,84,110–112] and inferences from our population genetic analyses. The logic and details can be found in Supplementary Note 6.

We conducted forward simulations in SLiM[35] under a Wright-Fisher model based on parameter estimates from the Fogo population to examine the probabilities of fixation of an adaptive variant (i.e., one that abolishes the vernalization requirement for flowering) taking into account the stochastic effects of drift. The selection coefficient ($s$) was set to 0.09273. Each simulation was run for a maximum of 6000 generations but was terminated earlier if a beneficial mutation arose and fixed. Mutation rate was set to $7 \times 10^{-9}$ and the probability of a beneficial mutation was set to match our estimate of $U_b = 1.54 \times 10^{-6}$ (Supplementary Note 6). We used three different plausible estimates for the degree of selfing (90%, 95 and 99%) based on estimates from *Arabidopsis* populations[113] and conducted 200 simulations for each case. From these, we calculated the proportion of runs where populations adapted, the proportions of potentially adaptive variants that are lost or fixed in all runs, and the times to fixation or loss.

**Statistical analyses**. For the comparison of climate variable distributions in Morocco and CVI, differences in the distributions were evaluated using two-tail Wilcoxon rank sum tests/Mann–Whitney U tests (hereafter MWW test) with the *wilcox.test()* function in R (https://github.com/HancockLab/CVI).

We computed the $d_{sel}/d_{neu}$ ratio and the distribution of fitness effects (DFE) with polyDfe v.2.0[109] for the two CVI island populations and Morocco. To estimate uncertainty around these parameters, we bootstrapped frequency spectra 500 times with polyDfe and calculated an empirical *p*-value for the $d_{sel}/d_{neu}$ ratio and the discretized DFE categories based on the bootstrapped data. The large variance in the bootstrapped data stems from the low number of variants segregating in CVI.

To assess fitness effects, we tested deme, habitat and deme x habitat interaction effects of Moroccan and CVI lines in the CVI and Moroccan-simulated environments. To correct for overdispersion, we employed a negative binomial transformation using the *glm.nb()* function from the package MASS v.7.3-51.4 in R (https://github.com/HancockLab/CVI).

To compute the proportion of private variants we counted the mutations that distinguish Cvi-0 from L*er*-0 and calculated the proportion which are private to Santo Antão and segregating there. This calculation was repeated for the whole genome, QTL and candidate genes. Because functional variants represent single mutations, in this case each variant was either fixed in CVI and denoted with 0% private, or segregating in Santo Antão and denoted with 100% private. For every functional category, we compared the rate of private variation to the genome-wide expectation (419466 variants differentiating Cvi-0 from L*er*-0, of which 3214 private ones), using a two-tailed Poisson test implemented in R (*poisson.test()*).

To assess the effects of the seven functional variants segregating in Santo Antão on fitness, we used forward-backward stepwise regression (i.e., sequential replacement) approach in a linear model framework using the R package caret v.6.0-86[114]. The significance of models was assessed based on the root mean squared error (RMSE) by 1000 bootstrap samples. To test whether the explanatory power of the seven functional variants was higher than randomly selected genomic variants, we resampled 2000 sets of seven randomly chosen variants from an LD-pruned genome (PLINK[104] command: <--indep-pairwise 50 10 0.1>) and conducted stepwise regression on each of these sets, exactly as we had done on the seven functional variants. We obtained an empirical *p*-value by comparing the observed R² to the resampled null distribution (https://github.com/HancockLab).

We tested for differences in the distributions of bolting time between CVI and Moroccan populations using two-tail MWW tests on the medians per genotype with the *wilcox.test()* function in R (https://github.com/HancockLab/CVI). 95% confidence intervals were calculated using function *ci()* implemented in the R package gmodels v.2.18.1[115].

To determine whether there was transgressive segregation in inter-island crosses, we tested each F2 population against their corresponding parental lines. Each parental line was grown in 12 replicates, except for Cvi-0 and F9-2

(4 replicates per lines), and the F2s had 488, 598, and 636, respectively for the crosses S5-10 x F13-8, Cvi-0 x F9-2, and S15-3 x F3-2. We used Dunnett's tests on each individual cross, using the *DunnettTest* function implemented in the R package *DescTools*[110] (https://github.com/HancockLab), and a Fisher's combined *p*-value test on the set of crosses, using the function *fisher.method* implemented in the R package *metaseqR*[111] (https://github.com/HancockLab).

We conducted genome-wide association studies (GWAS) using likelihood ratio tests in GEMMA[112] to test associations between markers and the median bolting time per natural line. Manhattan plots show *p*-values -$\log_{10}$ transformed on the y-axis.

We tested the difference in *FLC* expression and bolting time between genotypes with the Kruskal-Wallis method implemented in the R package *agricolae* (https://github.com/HancockLab). We applied the $2^{-\Delta\Delta Ct}$ (Applied Biosystems) on the median across three technical replicates per genotype.

For the *FLC* complementation test, we tested phenotypic complementation of F1 hybrids by comparing their phenotypic distributions to parental lines using the *wilcox.test()* function implemented in R (https://github.com/HancockLab), on four replicates of each of the parental lines and eight replicates of each F1 line. We tested for phenotypic complementation of Col-0 background F1 hybrids by comparing their phenotypic distribution to Col-0 *FRI*-Sf2 *flc-3* (*FRI*+*FLC*-) and Col-0 *FRI*-Sf2 (*FRI*+*FLC*+) using the *wilcox.test()* function implemented in R (https://github.com/HancockLab/CVI).

**Reporting summary**. Further information on research design is available in the Nature Research Reporting Summary linked to this article.

## Data availability

All data generated in this study are included in this article and its Supplementary Information files. The raw sequencing read data generated in this study have been deposited in the European Nucleotide Archive (ENA) under accession code PRJEB39079. In addition, previously published sequence data were used from ENA project ID PRJEB24044 and ENA project ID PRJNA273563. All sequences were aligned against the Arabidopsis TAIR reference assembly GCA_000001735.1. The genomic variant calls have been deposited in the European Variation Archive (EVA), under project accession number PRJEB44201. Source data are provided with this paper.

## Code availability

All code used in analyses and data visualization is available in the GitHub repository [https://github.com/HancockLab/CVI] and on Zenodo [https://doi.org/10.5281/zenodo.5844119][116].

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

## Acknowledgements

The authors thank Martin Koornneef, Nick Barton, Christian Brochmann, and George Coupland for valuable discussions and comments, and we thank Wolfram Lobin for sharing herbarium records. Logistical support in the field, field assistance and advice were provided by Natural Parks in Santo Antão and Fogo, Â. Moreno and S. Gomes at the Instituto Nacional de Investigação e Desenvolvimento Agrário (INIDA), Cape Verde, and Arlindo Martins. The project was supported by the Marie Curie CIG 304301, Vienna International Postdoctoral Program for Molecular Life Sciences (VIPS), NSF IRFP (1064766), Max Planck Society Funding, and ERC CVI_ADAPT 638810 to A.M.H., FWF DK W1225-B20 (A.F.), Laboratoire d'Excellence (LABEX) entitled TULIP (ANR-10-LABX-41) to F.R, DFG FOR 1078 to J.H. The funders had no role in study design, data collection and analysis, decision to publish, or preparation of the manuscript. All sample collection was made with appropriate field permits (PERMIT NUMBERS No.12/2012, 01/2015, 112/2018).

## Author contributions

Conceptualization: J.H., A.M.H; Methodology: A.F., C.N., A.M.H; Software: A.F.; Investigation, validation, and data curation: C.N., A.F., A.F.E., E.T., M.G., N.W., N.D., A.M.H.; Formal analysis: S.R., A.F., N.W., J.H., S.A., A.F.E., E.T., A.M.H.; Resources: H.D., C.N., E.T., P.J.F., A.F.E., A.F., F.R., A.M.H.; Writing-first draft: C.N., A.F., A.M.H.; Writing-reviewing and editing: all authors; Project administration: A.M.H.; Supervision and funding acquisition: M.K., F.R., J.H., and A.M.H.

## Funding

## Competing interests

The authors declare no competing interests.
