## [Peer Review File · Nature Communications]

Parallel reduction in flowering time from de novo mutations enabled evolutionary rescue in colonizing lineagesReviewers' Comments:

Reviewer #1:

Remarks to the Author:

This study reports an analysis of 335 *Arabidopsis thaliana* samples collected on two islands of Cape Verde. Genome-wide variation in these samples was compared to samples from continental Europe and Morocco to characterize and quantify genetic variation, estimate timing of CVI colonization and infer selection signatures. Simulated reciprocal transplant experiments were performed between CVI and Morocco to quantify fitness differences, and genetic variants underlying a fitness-related trait (i.e. early flowering) were identified and functionally validated. The results support models in which newly evolved mutations of large effect sizes underlie rapid adaptation and evolutionary rescue.

Most of the major claims made in this study are well supported and are potentially of broad interest. Below I address a couple of specific points.

Climate: it is emphasized that colonization of CVI by *A. thaliana* was associated with a (sudden) shift to arid climate (Abstract, line 6) and that AT on CVI live in 'a climatic extreme of the species range' (page 3, line 12). Both statements are not substantiated by data, analyses or adequate references. An analysis of the species' climatic niche with indication of conditions on CVI would be helpful. A direct comparison with conditions in Morocco, however, may not be adequate because it is unclear whether CVI was indeed colonized from Morocco (and the high Atlas mountains may represent another climatic extreme of the species' range) given that the geographic origin of the inferred ghost population is unknown.

At present, it remains open how strong the 'sudden and extreme environmental change' may have been for the colonizing lineage. Ancestral area and/or ancestral climate reconstruction may provide further insights.

The statement that 'this case can inform about more general mechanisms of adaptation after environmental change' requires justification. Colonization of CVI may potentially resemble an abrupt environmental change. However, environmental change associated with climate change is – for populations that are not colonizing geographically vastly isolated new habitats – not typically abrupt but instead gradual, raising the question what can be learned with respect to climate change adaptation in the great majority of cases from the situation studied here.

Colonization of CVI:

The statement that colonization of the two CV islands was associated with strong bottlenecks that 'wiped out pre-existing variation' appears to be in conflict with reported estimates of the numbers of colonizers. With > 40 inferred colonizers, some variation must have been retained. How can these aspects be reconciled? What are confidence intervals for the estimates of numbers of colonizers?

Evidence for adaptation

The population genetic signature of adaptation through dN/dS analysis is not convincing. As argued by Kryazhimskiy and Plotkin (2008) in PLoS Genetics in 2008, in population samples, 'it may be impossible to infer selection pressures' using dN/dS. Second, there is no convincing scenario – except the hypothesis of strong selection – provided that could explain the inferred massive genome-wide selection signature corresponding to dN/dS ratios of 1.7 and 2.2 on Fogo and Santo Antao. Third, it appears unlikely that time since colonization is sufficient to generate such a strong selection signature.

Local adaptation experiment: The authors have simulated a reciprocal transplant experiment in a climate chamber in which plants from CVI and Morocco were grown once under CVI climatic conditions and once under Moroccan conditions. Authors then discuss a 'home' and 'away' effect for fitness'. Given that these relevant results are available, it is preferable to report – in addition to the 'home vs away' contrast – also the 'local vs foreign' contrast, which is widely considered stronger evidence for

local adaptation (see Kawecki and Ebert, 2004 in Ecology Letters). In the methods, it would be relevant to indicate how relative fitness was calculated. The analysis of such experiments builds on testing for G x E effects for fitness, instead of the Mann-Whitney-Wilcoxon tests applied.

I find the conclusion drawn at the end of the study, namely that results 'imply that adaptation to increasing aridity – which is expected to be common under global climate change – is predictable' too strong. The studied situation may indeed resemble 'models of rapid adaptation and evolutionary rescue from new large effect mutations', as argued by the authors, but this does not imply that adaptation in other organisms in response to drastic environmental change necessarily follows such models. *Arabidopsis thaliana* is a primarily selfing species and adaptive processes in selfers can be expected to differ in important aspects from those in species with mixed mating systems or from outcrossers.

Reviewer #2:

Remarks to the Author:

In this study, the authors investigated the evolutionary and demographic history of populations of *Arabidopsis thaliana* near the edge of the species range on two Cape Verde islands using a variety of techniques and approaches. The results reveal the colonization history and show that the island populations experienced demographic decline until there were genetic changes that accelerated flowering time, allowing the populations to adapt to the arid conditions of the islands and rescuing them from potential local extinction. This work provides a case study in adaptive evolution and evolutionary rescue in colonizing populations that faced bottlenecks and novel climatic conditions.

I thought that this study was highly impressive, and the results novel and important. The authors used a large number of techniques and analyses on extensive data that provide an integrative and coherent investigation into this system. For example, the study not only identified the key genes associated with earlier flowering time, but also followed changes in these genes along with changes in population sizes over time, evaluated expression of these genes and functionally validated their phenotypic and fitness effects under simulated island and mainland conditions. This level of thoroughness was evident throughout the study and makes this a very strong body of research. In general, the studies were designed well, the analyses were appropriate, and the writing was clear and logically organized. The main conclusions of the study seem well supported. This study adds to the increasingly rich literature on the population genetics and evolutionary ecology of this model species. I think that this could be a classic and highly cited study. I did not have very major concerns, but I did have some comments that I think should be addressed to improve the paper.

The manuscript refers to local adaptation to Moroccan versus CVI conditions, but the study does not investigate local adaptation directly using reciprocal transplants. I think the approach is reasonable and practical, but this caveat should be included. In the study, the factors actually manipulated are climatic conditions rather than biotic factors or other unmanipulated abiotic factors, which could potentially also influence the degree of local adaptation to actual field conditions. Thus the wording should be changed in some cases to make it more clear that the study investigated the relative degree of adaptation to simulated climatic conditions in two different locations, rather than local adaptation to these locations.

The authors infer that alleles private to Santo Antão represented new mutations, but it is very difficult to distinguish between very rare variants that were present in the founding population compared to actual new mutations. I think this should be further explained.

I had some difficulty interpreting Fig. 5A. Are the bars representing different phenotypic categories showing all of the locations along the chromosome where there was a genetic association with this category? Is almost all of chromosome 5 associated with one of the light grey categories (I think this

is stress response)? Is the location of CRY2 on chromosome 1 associated with almost all phenotypic categories? The gray-scale rather than color made this difficult to read, and more information was needed to be able to interpret what this figure is really showing.

Fig. 7 seems to just compile information from Fig. 6i and Fig. 2C, so it is somewhat redundant. I realize that the point of this is to show that the timing of the population expansion coincided with the allele sweeping to fixation, but it seems like this point could be made by some rearrangements of the existing figures rather than adding a new figure. There are already a lot of figures. Also, this is just an opinion, but it would make more intuitive sense to me if the figures that have time on the x-axis read time going forward from left to right, rather than back in time from left to right.

Page 25, lines 8-10: "...connect mutations that occurred in parallel at FRI and FLC with adaptive divergence. This case provides a symmetric example of the importance of these two genes in adaptation to aridity." There are a couple of issues here. I would say that the phenotypic shifts to earlier flowering occurred in parallel on the two islands, but the fact that there were changes in two different genes indicates that the genetic basis of the phenotypic changes is non-parallel. I think that the way that "parallel" is used here is confusing given that parallel evolution has a very specific meaning. The genetic changes were simultaneous (roughly) but not parallel. They are also not symmetrical, and I would not call this a "symmetric example." Instead this appears to be two separate and related but independent cases of genetic changes leading to evolutionary rescue. The phenotypic changes were the same, and the genetic pathway was even the same, but the specific genetic changes were unique to each island. I think this could be clarified and the wording improved here.

Reference 16 is missing information.

Some of the references have each word in the title capitalized, while some capitalize only the first word.

Reviewer #3:

Remarks to the Author:

The Cvi accession of *A. thaliana* has long been known to be a special case, much diverged from other accessions. It has been frequently used in studies of functional variation. This ms now presents an interesting genetic analysis of colonization of (two) Cape Verde islands by *Arabidopsis thaliana*. The sequencing of more than 300 high quality, full genomes from the islands allows a detailed examination of the demography of colonization of first Santo Antão and then Fogo island, with estimates of N_e and timing of events. With these results available, there is an interesting genetic and functional analysis consistent with de novo mutations at two classical large-effect flowering time genes (one locus on each of the two islands colonized), rather than standing variation, having contributed to the fitness (and likely spread) of the species. The work can be of wide interest to evolutionary biologist.

The work is put into a context of island colonization at the broadest of terms in the introduction. The ms presents a very interesting case study, in this view, $n=1$. However, for instance, Brochmann et al. (1997) write that about one half of the species on Cape Verde islands were brought by humans (331 of 621). Definitely excluding this possibility would seem to merit explicit consideration and confirmation.

Here the data suggest that the actual origin population for *A. thaliana* is a ghost population, related to the Moroccan populations, but different. However, the model of evolution identified by their analysis identifies the existence of a ghost population as the origin of the colonizing population. How sure can we be that the mutations appeared indeed on the islands versus preexisting in the, unsampled, ghost population from mainland Africa? Could this eventuality be discussed when considering the SSWM vs WSSM alternatives?

The same authors have used similar analyses earlier to examine e.g. *A. thaliana* populations from Madeira, In Madeira, the colonization seem to have been older, but there seem to have been no

adaptation relating to flowering time. Some more comparison to this situation would be interesting.

"Intermediate" comments

-Given such a recent event, a hard sweep should be clearly observed in the two island populations. But this doesn't seem to be explored during the analyses, why?

After all the filtering based on *A. lyrata*, what actual numbers of nucleotide sites are included in the analyses?

Suppl. fig, 12 The expression data of FLC and the phenotypes are a bit puzzling. The FRI+FLC genotype seems to have similar levels of expression as the Santa Antao samples, but what are the phenotypes of these same genotypes. How come high FLC expression consistent with early flowering (as seems to be known for *Cvi-0* from earlier), a further comment would help

- L5p21: "Because we take the inferred change in population size into account in our estimate of the selection coefficient, s , our estimate is conservative and likely an underestimate. For example, inference under an assumption of constant population size leads to an estimate of $s = 23.56\%$." Maybe, but don't you risk as well overestimating the selection coefficient if your inference of the intensity of the population expansion is underestimated? This second possibility would seem a bigger and more relevant concern than the (unlikely) constant population size model for these populations.

Minor comments

Please check references: e.g.,

The reference to the original *Cvi-0* is given as Lobin (1983), rest of reference is missing.

- L4p27: reference in methods to the African strains is given as 8, which is a theory paper in *AmNat* by Uecker et al.

- dN / dS ratio within pop: some criticism against this practice see (Kryazhimskiy & Plotkin 2008)

- dN / dS: why not use a notation that reflects clearly you are using 4-fold and 0-fold degenerate positions?

- Fig1: "n=20 per cluster", perhaps a total number of samples used in this analysis would be nice to mention.

- Also Fig1: For the sake of helping readers understand better your results on Fig1b, why not display the individual clusters in the Eurasian group in different colors? Especially the Ir and Inr would be an interesting group to outline. This would help readers compare this tree with previous results eg Fulgione et al. 2018

Supp Fig. 13 perhaps you could consider the color scheme, can be a bit confusing at first (comment by embarrassed reviewer)

- p3 L5: "ameliorate risk"  ameliorate risk prediction

- p12 L7: could it be useful to clarify the origin of *Ler-0*?

- L6p26: *A. thaliana* not in italic

Reviewer #4:

Remarks to the Author:

The authors study adaptation to dry climate in *A. thaliana* on the Cape Verde Islands. They show that two different loss of function mutations on two different islands (Santo Antao and Fogo) both lead to earlier flowering time by allowing flowering without a previous cold period. Evidence suggests that both mutations appeared de novo in populations of small effective size, corresponding to a strong-selection-weak-mutation regime.

This is an impressive study telling a remarkably complete story drawing on a wide range of evidence and written in a very clear style. Unfortunately, I am not familiar with the details of the molecular and computational tools employed, so I cannot really comment on the

methods. I will therefore focus on the proposed evolutionary scenario.

The authors argue that the S. Antao population was founded between 7-5000 years ago from an unknown "ghost" population which had split some 40000 years ago from the Moroccan lineage. About 2000 years later (3-5 thousand years ago), the S. Antao population gave rise to the Fogo population. Then both populations acquired independent adaptations. On S. Antao a mutation of the FRI gene, that advances flowering time by about 35 days, appeared and rose to high frequency in the majority of local populations. Apparently, this mutation arose after the colonization of Fogo. On Fogo, another mutation in the FLC, which advances flowering time by 27 days, arose "very soon" after the establishment of the population and subsequently went to fixation. At about the same time, the Fogo population started to expand from an initial, 1000 year long bottleneck.

Estimates of the effective population size and the mutational target size suggest that the mean waiting time for the appearance of the FLC mutation is around 3000 generations (somewhat less for the FRI mutation). Estimated selection coefficients are between 9 and 23 percent for the FRI mutation and around 4 percent for the FLC mutation. Unless selfing rate is extremely high, fixation probabilities are of the same order as the selection coefficients, and fixation times are short (around 50 generations). So, while the mean waiting time for a successful mutation might well be on the order of 10000 generations or higher, such mutations apparently appeared considerably faster in the Cape Verde populations, and these fortuitous events might have contributed to long-term population survival.

I don't have any serious criticisms of this scenario, just a few questions/comments/clarifications:

- Given that both populations survived for about 1000 generation before the appearance of the two mutations, I find the term "evolutionary rescue" a bit strained.
- I might have missed it, but are there any data about the phenotype and relative fitness of S. Antao individuals that do not carry the FRI mutation?
- Can one say something about the ratio of effective to census population size?
- page 10, 10: How can the results from the fitness assays (8- and 16-fold advantage of CVI forms) be reconciled with estimates of selection coefficients ("only" 0.09-0.23)? See also 140 increase in seed number for the FRI mutation (page 19, 10).

Minor comments:

- 9, 18: What do the negative percentages in parenthesis mean?
- 9, 19: DFE of what? segregating variants?
- 10, 13: say something about how much higher fitness of local plants was in Morocco (the difference is much weaker than on CVI)
- 12, 22: enrichment with respect to what?
- 24, 18: what do you mean by symmetric?
- 29, 15: something seems to be missing before "and the R package ape"

- 31, 5: I'm a bit confused by the "upper" and "lower" bound to the colonization time. Which one is closer to the present?

EDITORIAL COMMENTS

Thank you again for submitting your manuscript "Parallel reduction in flowering time enabled evolutionary rescue and establishment in a colonizing *Arabidopsis* lineage" to Nature Communications. We have now received reports from 4 reviewers and, on the basis of their comments, we have decided to invite a revision of your work for further consideration in our journal. Your revision should address all the points raised by our reviewers (see their reports below).

When resubmitting, you must provide a point-by-point response to the reviewers' comments. Please show all changes in the manuscript text file with track changes or colour highlighting. If you are unable to address specific reviewer requests or find any points invalid, please explain why in the point-by-point response.

We thank the editor and the four reviewers for the time spent evaluating our manuscript. The comments and suggestions were extremely helpful for us to identify areas where we could improve our manuscript and we are excited to share the revised version.

Below, we provide an overview of the new analyses we conducted and the most significant changes we made. Then we provide point-by-point responses to the reviews.

Changes are summarized below:

General modifications based on reviewer comments:

Some of the reviewer comments were particularly helpful in that they forced us to think more about the broader context of our results and relevance of our findings. As a result, we made some minor changes to the Title, Abstract and Introduction to ensure these messages were clear. These included the phylogenetic structure of the populations, convergent versus parallel genetic changes and the relevance of new mutations versus standing variation in adaptation in CVI. All changes to the manuscript are marked in red.

Changes in figures and tables:

We added one new figure to the main text (Fig. 4), one new figure to Supplementary Results (Supp. Fig. 2), and we moved a figure from the main text to Supplementary Results (Fig. 7 -> Supp. Fig. 14). The new Supplementary Figure 2 shows the distributions of 10 representative climate variables at *A. thaliana* collection locations and Supplementary Table 2 shows the results for all climate variables analyzed. The new Figure 4 summarizes the results of niche modelling for the Moroccan *A. thaliana* population (outgroup to CVI) and projection onto the Cape Verde habitat. Supplementary Table 5 provides the percent contribution and percent importance for variables included in the niche model. We also inserted a new panel into Fig. 5 that shows a schematic of the phylogeny underlying the dN/dS analysis to help explain the rationale for the dN/dS approach.

Based on a suggestion from Reviewer 2, we moved the previous Figure 7, which shows the congruence between the initial population size increase in Fogo and the fixation of *FLC*, from the main text to the Supplement, so that this is now Supplementary Figure 14.

As requested based on Nature Communications guidelines, we created (excel) files that contain source data for figures. As a result, we removed many of the previous Supplementary Tables that simply provided the source data for figures and moved these to the source data files so that the Supplementary Table names have changed.

Other major changes fall into the following categories:

1. Climate in Cape Verde relative to Eurasia and Morocco

Two reviewers asked for more information about the climate in Cape Verde relative to that recorded at Moroccan and Eurasian sites to show evidence for a shift in climate upon colonization. In the revised manuscript, we compare these distributions directly and also conduct niche modeling to retrace what might have occurred at colonization. For this work, we involved an additional lab member, Shifa Ansari, who is now listed as a co-author on the manuscript.

We extracted climate data for the collection locations of *A. thaliana* samples from Eurasia (1001 Genomes Consortium 2016), Morocco (64 samples from 20 populations (Brennan et al. 2014; Durvasula et al. 2017) and Cape Verde (samples sequenced in this study). These data included 19 Bioclim variables from the WorldClim Project as well as aridity index (Precipitation/Potential evapotranspiration) and the length of the growing season. We focused our analysis on actual samples collected rather than including herbarium samples because we knew that some herbarium sample location information in the GBIF database for CVI samples was incorrect and its inclusion would have resulted in climate distributions that would have been anti-conservative (these locations have more extreme climate in terms of aridity and temperature than locations where *Arabidopsis* is actually found in CVI). Similarly, in Morocco, sites that did not overlap with known collection locations had 'fuzzy location' information and were also not reliable and were thus not added to the data set. Information about the data used can be found in the Methods (page 29) and Supplementary Methods (pages 3-4).

We conducted Mann-Whitney Wilcoxon (MWW) tests to compare climates across the species distributions in Eurasia and Morocco to those in CVI. The CVI climate is quite different to Eurasia and Morocco at the locations where *Arabidopsis* has been found. Nearly all comparisons were significant with a conservative Bonferroni correction for multiple tests. The revised version of the manuscript includes a table (Supp. Table 2) with climate medians in the three regions and results of MWW tests as well as a figure that shows plots of the distributions of 10 representative variables (Supp. Fig. 2). The results are described in the manuscript main text on page 4, lines 12-19 and analysis is described on page 38.

We also used niche modeling in Morocco to ask which climatic factors limit the distribution there and to assess how suitable the CVI environment would have been to a colonizing population that has evolved to fit the Moroccan climatic niche. The methods are described in Supplementary Methods, pages 11-12 and results are described in the main text on page 11, lines 1-19. For this, we used Maxent to produce a climatic niche model for the Moroccan population using the set of variables we used in the MWW analysis described above (Fig. 4a-b). These were pruned based on correlation among variables to avoid over-fitting. We then used these models to predict where in CVI the Moroccan populations would be expected to occur based on environmental suitability. We tried several approaches to pruning the variables and all produced the same result in the analysis of suitability of the CVI environment to a Moroccan population: that there is no predicted suitable region for Moroccan populations in CVI.

The main factors limiting the distribution in Morocco were growing season length and temperature (max T in the warmest month and min T in the coldest month). This is summarized in Supp Table 5. Although we found no locations that were predicted to be suitable for *A. thaliana* based on the Moroccan modeling (Fig. 4c), we did find that the CVI locations predicted to be most similar to the Moroccan range were those where *Arabidopsis* is actually found (Fig. 4d). That is, the most similar regions in CVI were found in Santo Antão and Fogo and similarity was highest where we actually found *Arabidopsis* populations. The finding that no regions were 'suitable' based on this analysis is due to the strong multivariate climatic differentiation between Morocco and CVI. The results are consistent with the evidence of strong adaptive evolution in CVI (based on dN/dS, fitness differential in the simulated CVI environment and functional variants from QTL). The niche modeling results are further consistent with the inference that the *FRI* and *FLC* loss of function events were important for

Arabidopsis to establish on the islands and escape ultimate extinction. These results are described on page 11, lines 1-16.

2. Phylogenetic structure of the populations

Several comments from the reviewers gave us the impression that we were not clear enough in some of our descriptions of the colonization history and the implications w.r.t. the phylogenetic separation of the two Cape Verde Islands compared to mainland Morocco and to each other.

The near-complete lack of shared segregating variation between the CVI and Morocco and between CV islands is analogous to the level of divergence between pairs of species in the *Arabidopsis* genus. Actually, it is stronger than what has been found previously for different *Arabidopsis* species; see (Novikova et al. 2016). As a result, both the Santo Antão and Fogo populations would be considered separate species based on a phylogenetic species concept. Based on several concerns that arose in the reviews, we realized that this point is really crucial for the reader to understand the logic of the analyses and results and thus we now emphasize it more in the text (page 2, lines 5-7, page 5, lines 3-4, page 5, lines 7-9, page 17, lines 13-16).

a. Estimating the probability that variation derives from new mutation that arose on the islands versus standing variation that was present in the founding population

Reviewers 1, 2 and 3 suggested that we needed more information about the probability that variants segregating in Santo Antão were due to new mutations that arose after colonization versus standing variation from the ghost population. Although it was not necessarily explicitly stated, it seemed to us that the motivations for this question were mainly related to addressing whether adaptation occurred through new or pre-existing variation, which is a central question in evolutionary biology and therefore an important point to clarify.

We have now conducted coalescent simulations to estimate the probability that mutations segregating in CVI derive from standing variation in the founding population. We based our simulations of the Santo Antão population on our inference of N_e over time (Fig. 2c). We find that the expected proportion of variants currently segregating in Santo Antão that were introduced from segregating variation in the founder ('ghost') population is exceedingly low (mean: 0.017%, 95% CI: 0.006-0.03%). That is, on average, only 1.7 out of 10,000 variants currently segregating in Santo Antão would be expected to have been segregating in the colonizing ('ghost') population. The total number of variants segregating on this island is 18,122, so that would translate to only approx. 3 variants that still segregate on the island. These simulations assumed neutrality, but selection would only increase the rate of loss or fixation of these variants (depending on the direction of selection) so that selected variants would be even less likely to continue to segregate in the population. Further, there is additional evidence that the specific variants of greatest interest here, *FRI 232X* and *FLC 3X*, arose in CVI. *FLC 3X* is not found in Santo Antão at all, and *FRI 232X* is absent from the Santo Antão sub-population that appears to represent the most 'relictual' population – i.e., the Cova sub-population, which split off earliest in our intra-island reconstructions (Supplementary Fig. 7). Further, the coalescence time we inferred for these variants is much more recent than the estimated founding event.

The probability that variants segregating in CVI derive from standing variation in the founding population is now discussed in the manuscript on page 6 (lines 5-9).

b. dN/dS analysis

Two reviewers raised issues regarding applying the dN/dS statistic to population-level data based on a paper by Kryazhimskiy and Plotkin. The concern discussed in this reference is that if individuals from populations that exchange genetic material are used for calculating dN/dS (by comparing the two populations) then polymorphism could be counted as divergence, the theoretical basis of dN/dS would be violated, and the result could therefore be misinterpreted.

Note that the entire argument of Kryazhimskiy and Plotkin centers around using individual sequences to represent populations and finds that it is problematic to compare these individual sequences to each other. In our analyses of continental populations we were comparing each directly to the *A. lyrata* outgroup, not to other *A. thaliana* populations. While we compare CVI 'populations' to Morocco, these actually represent different species from the phylogenetic perspective (with less than 1% shared variation). Therefore, our analyses as presented in the previous version do not suffer from the potential pitfalls described by Kryazhimskiy and Plotkin.

However, given that two of the four reviewers were concerned about this, we wanted to make some changes to ensure that these points are clear in the final version of the manuscript. To this end: (1) we simplified the set of populations analyzed and (2) we added a figure panel (Fig. 5a) that shows a schematic of the structure of the phylogeny analyzed and additional text to explain the phylogenetic structure of the Cape Verde populations and the specific relationship of our analysis to that which concerned Kryazhimskiy and Plotkin.

Regarding point 1: since the only calculations really relevant to our analysis are those for Morocco, the CVI divergence branch and variation specific to each island, we reduced our reported dN/dS (now 'd_{sel}/d_{neu}') results to these analyses. More specifically, our analyses focus on (i) the divergence of Morocco compared to *A. lyrata*, (ii) the shared branch of variants that are differentiated from Morocco but fixed in CVI ('CVI divergence') compared to Morocco, and the branches leading to fixed variants specific to either (iii) Fogo or (iv) Santo Antão (both in comparison to Morocco). See page 13, lines 6-22 and page 14, line 1 of the main text for the discussion of this.

We should also note here that we have changed the notation 'dN/dS' to d_{sel}/d_{neu}' in response to a comment by Reviewer 3. This is because the statistic we calculate only includes the extremes: 0-fold and 4-fold degenerate sites rather than the complete set of possible NS sites. The interpretation of the statistic is the same as for dN/dS; we opted for this approach because estimating the expected number of changes is less accurate for partially degenerate sites (2-fold and 3-fold degenerate sites) (discussed in the MEGA4 manual (<https://www.megasoftware.net/mega4/mega4.pdf>) on pages 130-136). This is mentioned in the main text (page 13, lines 10-12) and discussed on page 12 of Supplementary Methods.

3. Analysis of the fitness experiment

Reviewer comments regarding the fitness experiment were very helpful. We have changed this section to use a GLM framework to include GxE effects. Since the distributions of seed counts (fitness) contain a fair number of zeroes, especially in the case of Moroccans in CVI we used a negative binomial transformation in the GLM). The change in this analysis approach resulted in a change in Figure panel 5d and changes in the main manuscript on page 14, lines 16-22 and page 15, lines 1-9. Methods are provided on page 39, in the Statistical Analyses section. Also in response to a comment from Reviewer 2, we added caveats about the aspects of the natural environments (biotic, edaphic) that would not be reflected in the simulated conditions and the fitness experiment results.

Other changes, not requested by reviewers:

We noticed that the reporting of diversity in the islands was confusing because we provided only a combined measure of the fold reduction whereas we provided island-specific thetas. To rectify this, we changed the text on page 4 lines 21-22. The complete sentence now reads (underline denotes changed text):

Diversity within islands is 73.3- and 62.3-fold reduced compared to the continent (θ_w (Santo Antão) = 7.59×10^{-5} , θ_w (Fogo) = 8.93×10^{-5} , θ_w (Morocco) = 5.56×10^{-3} ; Supplementary Table 3) and there is almost no shared variation between the islands and Morocco or between the two Cape Verde Islands (Fig. 2a-b).

REVIEWER COMMENTS

Reviewer #1 (Remarks to the Author):

This study reports an analysis of 335 *Arabidopsis thaliana* samples collected on two islands of Cape Verde. Genome-wide variation in these samples was compared to samples from continental Europe and Morocco to characterize and quantify genetic variation, estimate timing of CVI colonization and infer selection signatures. Simulated reciprocal transplant experiments were performed between CVI and Morocco to quantify fitness differences, and genetic variants underlying a fitness-related trait (i.e. early flowering) were identified and functionally validated. The results support models in which newly evolved mutations of large effect sizes underlie rapid adaptation and evolutionary rescue.

Most of the major claims made in this study are well supported and are potentially of broad interest.

We thank the reviewer for the positive comments.

Below I address a couple of specific points.

Climate: it is emphasized that colonization of CVI by *A. thaliana* was associated with a (sudden) shift to arid climate (Abstract, line 6) and that AT on CVI live in 'a climatic extreme of the species range' (page 3, line 12). Both statements are not substantiated by data, analyses or adequate references. An analysis of the species' climatic niche with indication of conditions on CVI would be helpful. A direct comparison with conditions in Morocco, however, may not be adequate because it is unclear whether CVI was indeed colonized from Morocco (and the high Atlas mountains may represent another climatic extreme of the species' range) given that the geographic origin of the inferred ghost population is unknown.

At present, it remains open how strong the 'sudden and extreme environmental change' may have been for the colonizing lineage. Ancestral area and/or ancestral climate reconstruction may be provide further insights.

We found this comment really helpful. Climatic divergence was actually the major impetus for the project because Cvi-0 was such a strong climatic and genetic outlier, I found it caused problems in a linear mixed model analysis of climate associations (in the process of working on (Hancock et al. 2011)). Therefore, it was a silly oversight not to include any quantification of the climate differences here. We have now conducted direct comparisons of the climate with CVI and niche modeling in the context of the inferred colonization event. The new analysis results are presented in Figure 4, Supp Figure 2, Supp. Tables 2, 4-5 and are described above in the overview of this document on page 2.

Briefly, we compared the climate variable distributions across collection locations in CVI to those in Eurasia and Morocco for 21 bioclimatic variables (19 from the Bioclim project and 2 that we added because they were more closely related to focal points of the paper (i.e., aridity index and growing season length). See page 4, lines 12-19. Then, we conducted niche modeling in Morocco with the rationale that this is the closest present-day representative to the CVI outgroup. We used this model to identify the most similar regions in CVI and to ask if these would be predicted to be suitable to *A. thaliana* based on the climatic factors that limit the distribution in Morocco. These results are described on page 11, lines 1-18.

The reviewer suggested that it would be nice to also use historical climate information, but, as mentioned in the review itself, this is not really feasible. There are two issues: 1) we cannot really be

certain what distribution *A. thaliana* had in Africa 7-10 kya, when we estimate colonization to have occurred, and (2) there are severe limitations in the quality and resolution of the climatic inferences during that time frame.

Although we can never be certain about past events, it seems as if the most parsimonious explanation would be that although the African distribution of *A. thaliana* might have been more extensive than it is now and this suitable niche would have shrunk after the end of the last pluvial, the climate envelope would have likely stayed the same or be similar. Of course other possibilities could be imagined, but given that there is no particular evidence to the contrary, it seems to be a reasonable assumption.

Given the stark differences in present-day Moroccan and CVI environments, we infer that the colonization event was associated with a sudden shift in environment. This is further supported by results from other analyses in our manuscript. It is consistent with our findings of strong evidence of positive selection on the branches leading to the island populations (dN/dS and alpha estimates), the QTL analysis. This is further bolstered by the fact that these adaptations derive from variants that we can be 99.9% certain arose within Cape Verde (please see discussion on page 2 of this document for this quantification). The inferences are further consistent with our finding that *FRI* and *FLC* loss of function mutations, which also are private to individual Cape Verde islands have strong positive impacts on fitness and would have likely been necessary for populations to thrive in CVI.

In summary, we believe that the new climate analyses strengthen the paper and further strengthen some of the other conclusions we make in the manuscript.

The statement that ‘this case can inform about more general mechanisms of adaptation after environmental change’ requires justification. Colonization of CVI may potentially resemble an abrupt environmental change. However, environmental change associated with climate change is – for populations that are not colonizing geographically vastly isolated new habitats - not typically abrupt but instead gradual, raising the question what can be learned with respect to climate change adaptation in the great majority of cases from the situation studied here.

We agree that the CVI case represents an abrupt change, but still believe it is a useful model for processes that occur in mainland populations. One benefit of working in island system is they can act as simplified microcosms, where evolutionary history may be clearer than on continents where migrations and secondary contact often obscure such patterns. In response to this comment, we added ‘rapid’ as a qualifier to the sentence on page 4, line 6, which now says that the CVI “case can inform us about ... adaptation after rapid environmental change”.

We do not explicitly refer to ‘climate change’ but rather mention ‘environmental change’ more broadly so it doesn’t seem that it necessarily has to be explicitly relevant for climate change. However, based on recent events, it does seem that climate change may actually cause surprisingly rapid increases in temperature and changes in precipitation patterns (Tabari 2020; Schiermeier 2021), due to compounding effects of drought and release of methane stores (Dessandier et al. 2021; Walter et al. 2006; Steinbach et al. 2021; Schuur et al. 2015). Although the effects will differ across space, there will be high-risk areas where climatic factors are expected to be outside the long-term ranges (O’Neill et al. 2016; Tollefson 2020; Konapala et al. 2020), and this is expected to result in increased extinction risk (Sperle and Bruelheide 2021; Urban 2015; Intergovernmental Science-Policy Platform on Biodiversity and Ecosystem Services, IPBES 2019). In many cases, migration is not expected to occur fast enough to supply the variation needed to aid adaptation, which could result in additional local extinction events.

In this study, we specifically found evidence for a simple and clear pattern of parallel adaptation for reduced time to flowering and concomitant increases in fitness based on genetic convergence at the two loci (*FRI* and *FLC*). Similar loss and reduction in function of *FRI* and *FLC* in continental

populations have been found in *A. thaliana* and other species and it has been suggested these may result from adaptation to shorter growing seasons. However, this is exceedingly difficult to prove amidst the complexity in continental systems. Thus, the results from Cape Verde seem to provide important evidence regarding the role of similar mutations in continental population of *A. thaliana* and other species.

So, as was the case for the generalizations that Darwin and Wallace made based on what they observed in island systems, we believe that this is similarly a case where the simplicity of the island system can provide a clear view of processes that are also very relevant in continental populations.

Colonization of CVI:

The statement that colonization of the two CV islands was associated with strong bottlenecks that 'wiped out pre-existing variation' appears to be in conflict with reported estimates of the numbers of colonizers. With > 40 inferred colonizers, some variation must have been retained. How can these aspects be reconciled?

What are confidence intervals for the estimates of numbers of colonizers?

To quantitatively assess the effect of the bottleneck on pre-existing variation, we conducted genome-wide simulations using the N_e trajectory in Santo Antão from RELATE to assess the amount of variation from the founding population that would be expected to persist to the present day. We found that on average across 100 simulations 0.017% of variants segregating in Santo Antão are expected to be due to segregating variation in the founder, which equates to about 3 variants out of the total approx. 18000 in Santo Antão. These support the statement that this bottleneck eliminated pre-existing variation. The simulation results are now discussed in the revised manuscript on page 6, lines 3-9.

We have also now estimated confidence intervals for the number of colonizers. To do this, we applied a resampling approach to estimate the variance across trees inferred from across the genome using the COLATE package (<https://github.com/leospeidel/Colate>). We have added this information to Figure 2c and the associated source data file. Slight changes in the estimated initial population sizes in Santo Antão and Fogo (now 40 and 48, respectively) required a few other minor changes such as in the SSWM section on page 26 of the main text and on pages 45-46.

Evidence for adaptation

The population genetic signature of adaptation through dN/dS analysis is not convincing. As argued by Kryazhimskiy and Plotkin (2008) in PLoS Genetics in 2008, in population samples, 'it may be impossible to infer selection pressures' using dN/dS.

Since we are comparing each population to its phylogenetically distinct ancestor, the issues raised by Kryazhimskiy and Plotkin, who looked at the case where pairwise comparisons are made between *individuals* of the *same* phylogenetic species, are not relevant here. An important point here is that the level of shared variation between each CVI population and Morocco is very low (i.e., less than that observed for other species pairs in the *Arabidopsis* genus), these can effectively be treated as separate species.

However, since this issue was raised by two different reviewers, we thought it was really important to ensure clarity more generally for the expected readers of the manuscript. Therefore, we simplified the analysis and the results presented and provided more rationale and explicit explanation of how the results differ from the model discussed in Kryazhimskiy and Plotkin. We also made clearer statements about the phylogenetic separation of the CVI islands from Morocco and from each other and added a schematic of the relationships (text on page 5, lines 3-9, Figure 5a), which help readers to understand in the context of our dN/dS analysis (now termed dsel/dneu based on a comment from

Reviewer 3). Please see pages 13-14 for the changes related to dN/dS and please note that we changed the notation of the test statistic to denote that we do not include all NS mutations in our analysis. Our reasoning is discussed in the manuscript but briefly: expected changes are more straightforward to tally for 0- and 4-fold degenerate sites and therefore the results and conclusions based on these are more robust than the other types of sites.

For more discussion of this, please see our detailed discussion in the introductory statements on pages 3-4 of this document.

Second, there is no convincing scenario – except the hypothesis of strong selection - provided that could explain the inferred massive genome-wide selection signature corresponding to dN/dS ratios of 1.7 and 2.2 on Fogo and Santo Antao.

This comment was a little bit confusing to us. The dN/dS statistic is scaled such that neutral mutation (lack of purifying or positive selection) would result in dN/dS = 1.0. That is, in the calculation of the statistic, the actual numbers of observed nonsynonymous and synonymous substitutions are divided by those expected based on the nucleotide composition of the genome. As a result, a value of 1.0 is expected if all loci in the genome are evolving neutrally (i.e., a complete lack of positive or purifying selection). A significant departure above 1.0 is evidence of adaptive substitution, while a significant departure below 1.0 is evidence of purifying selection.

However, we could have stated explicitly that multiple forces are likely to be acting simultaneously across the genome and now we have added some information about caveats. A reduction in purifying selection is one force that could be acting to help push dN/dS upwards from 0.0 towards 1.0. It is very unlikely that a complete loss of purifying selection would result in a viable lineage so that this could only really explain part of this upward force. Positive selection is needed to explain a value of dN/dS above 1.0.

This section is largely rewritten to improve clarity (pages 13-14). We now mention that relaxed purifying selection could be responsible for some of the increase in dN/dS (now termed $d_{\text{sel}}/d_{\text{neu}}$) ratio up towards 1.0, and likely contributes to the signals within each island. We also simplified the analysis to only include Morocco, the divergence branch which contains a mix of signals from before (ghost) and after colonization, and each island and this is reflected in the new Figure panels 5a-b.

A little bit later in this section (in the discussion of alpha) we also added references to two papers that show that estimates of alpha may be inflated under a history of increasing population size, which we thought was also related to the reviewer's point (Rousselle et al. 2018; Eyre-Walker 2002).

Third, it appears unlikely that time since colonization is sufficient to generate such a strong selection signature.

We are not completely sure what the reviewer means by this. We think it may help to provide some context. In experimental evolution experiments with microbes, the strongest signals are often observed early after the start of the experiment (see the large elevation of non-synonymous mutations in the first 5000 generations in the long-term evolution experiment, Fig 4b, in Tenaillon et al, Nature volume 536, pages 165–170 (2016)). This is consistent with the idea that after landing in a harsh, new environment, a population is far from the local adaptive optimum, and the early steps in an adaptive walk are likely to consist of mutations with large beneficial effects resulting in a strong signature of selection. This is also consistent with the Orr-Fisher geometric model of adaptation.

The shallow timescale of CVI evolution results in a relatively low number of fixed derived mutations in each branch, which is reflected in the large confidence intervals for the estimate of dN/dS, which are clearly displayed in Fig. 4a.

Local adaptation experiment: The authors have simulated a reciprocal transplant experiment in a climate chamber in which plants from CVI and Morocco were grown once under CVI climatic conditions and once under Moroccan conditions. Authors then discuss a “‘home’ and ‘away’ effect for fitness’. Given that these relevant results are available, it is preferable to report – in addition to the ‘home vs away’ contrast – also the ‘local vs foreign’ contrast, which is widely considered stronger evidence for local adaptation (see Kawecki and Ebert, 2004 in Ecology Letters). In the methods, it would be relevant to indicate how relative fitness was calculated. The analysis of such experiments builds on testing for G x E effects for fitness, instead of the Mann-Whitney-Wilcoxon tests applied.

We thank the reviewer for this comment and for pointing out the relevance of the local vs foreign contrast. We have now reanalyzed these data to incorporate GxE effects. This is described in the introductory section of this response. Based on this analysis and the discussion around Fig. 1 in Kawecki and Ebert 2001, the effect we find should in fact be described as a ‘local’ vs ‘foreign’ effect rather than a ‘home’ and ‘away’ effect. More specifically, we find higher absolute fitness for Moroccans and Cape Verdeans in the Moroccan simulated environment together with higher relative fitness for CVI compared to Morocco in CVI conditions and for Moroccans compared to CVI in Moroccan conditions. Relevant changes in the manuscript are shown in red on page 14-15. `more details related to this can be found in the introductory text of this document on page 4.

I find the conclusion drawn at the end of the study, namely that results ‘imply that adaptation to increasing aridity – which is expected to be common under global climate change – is predictable’ too strong. The studied situation may indeed resemble ‘models of rapid adaptation and evolutionary rescue from new large effect mutations’, as argued by the authors, but this does not imply that adaptation in other organisms in response to drastic environmental change necessarily follows such models. *Arabidopsis thaliana* is a primarily selfing species and adaptive processes in selfers can be expected to differ in important aspects from those in species with mixed mating systems or from outcrossers.

We toned down the statement referenced here by changing ‘imply’ to ‘suggest’ and also added a reference to the reduction in growing season (page 28, lines 5-6), which is perhaps the more specifically relevant variable based on our analysis of the climatic differentiation and niche modeling.

However, we do not know of any particular reason why this statement should be qualified to be specific to selfing species. As we cite in our Discussion section, observations of loss of *FRI* and *FLC* function across a variety of species (selfing and outcrossing) suggest that these species are not so different in their responses to selection. Further, climatic changes like the ones we discuss here have been suggested to underlie these loss of function events. The simple history and convergence of two different paths to reduce flowering time that we find here therefore helps to strengthen the argument that selection due to climatic factors like aridity and growing season length may be responsible for the patterns observed in diverse species.

Reviewer #2 (Remarks to the Author):

In this study, the authors investigated the evolutionary and demographic history of populations of *Arabidopsis thaliana* near the edge of the species range on two Cape Verde islands using a variety of techniques and approaches. The results reveal the colonization history and show that the island populations experienced demographic decline until there were genetic changes that accelerated flowering time, allowing the populations to adapt to the arid conditions of the islands

and rescuing them from potential local extinction. This work provides a case study in adaptive evolution and evolutionary rescue in colonizing populations that faced bottlenecks and novel climatic conditions.

I thought that this study was highly impressive, and the results novel and important. The authors used a large number of techniques and analyses on extensive data that provide an integrative and coherent investigation into this system. For example, the study not only identified the key genes associated with earlier flowering time, but also followed changes in these genes along with changes in population sizes over time, evaluated expression of these genes and functionally validated their phenotypic and fitness effects under simulated island and mainland conditions. This level of thoroughness was evident throughout the study and makes this a very strong body of research. In general, the studies were designed well, the analyses were appropriate, and the writing was clear and logically organized. The main conclusions of the study seem well supported. This study adds to the increasingly rich literature on the population genetics and evolutionary ecology of this model species. I think that this could be a classic and highly cited study. I did not have very major concerns, but I did have some comments that I think should be addressed to improve the paper.

We thank the reviewer for these comments and for their positive feedback.

The manuscript refers to local adaptation to Moroccan versus CVI conditions, but the study does not investigate local adaptation directly using reciprocal transplants. I think the approach is reasonable and practical, but this caveat should be included. In the study, the factors actually manipulated are climatic conditions rather than biotic factors or other unmanipulated abiotic factors, which could potentially also influence the degree of local adaptation to actual field conditions. Thus the wording should be changed in some cases to make it more clear that the study investigated the relative degree of adaptation to simulated climatic conditions in two different locations, rather than local adaptation to these locations.

We changed the wording of this part of the manuscript to clarify that our experiment tested adaptation to specific simulated conditions and in particular not to edaphic or biotic factors.

We added statements in the results and conclusions mentioning these caveats. In the Results section (page 14, lines 21-22) we added:

These experiments aimed to examine the fitness effects of climatic factors that differentiate CVI and Morocco and would not capture abiotic or edaphic factors important for fitness.

And in the Conclusions (page 27, lines 11-16) we added the following:

Our population genetic analyses (Fig. 5a-b) and investigation of patterns at known functional loci (Fig. 6c) further suggest that adaptation in Cape Verde was multivariate and involved many loci and traits. Some of these would be reflected in fitness differentials in the simulated CVI and Moroccan environments. But others – in particular differences in abiotic and edaphic factors – would not be captured in our simulated conditions. Future work in these new Arabidopsis island lineages will be necessary to better characterize the multivariate history of adaptation here.

The authors infer that alleles private to Santo Antão represented new mutations, but it is very difficult to distinguish between very rare variants that were present in the founding population compared to actual new mutations. I think this should be further explained.

This is similar to comments from Reviewers 1 and 3 and is discussed in the introductory text to the Reviewers (on page 3 of this document). The severe colonization bottleneck coupled with slow

population expansion on Santo Antão would have removed nearly all of the variation segregating in the ghost population. To quantify the effect of the bottleneck, we conducted simulations based on the reconstructed N_e trajectory in Santo Antão and asked what proportion of the variation segregating in Santo Antão today can be traced back to segregating variation at the time of the split. We modelled the split based on the inference from haplotype coalescences within Santo Antão (see Fig. 2c and associated source data for Santo Antão), setting the effective population size of the in the ghost to 10k based on historical N_e in Morocco (see (Durvasula et al. 2017)). Using 100 whole genome simulations, we found that 0.017% (95%CI: 0.006-0.03%) of the variants that currently segregate in the simulated Santo Antão population result from variation present in the ghost ancestor. We now discuss this in the manuscript (page 6, lines 5-9).

I had some difficulty interpreting Fig. 5A. Are the bars representing different phenotypic categories showing all of the locations along the chromosome where there was a genetic association with this category? Is almost all of chromosome 5 associated with one of the light grey categories (I think this is stress response)? Is the location of *CRY2* on chromosome 1 associated with almost all phenotypic categories? The gray-scale rather than color made this difficult to read, and more information was needed to be able to interpret what this figure is really showing.

We have modified this figure to improve information content and clarity. Each bar (or segment) along the chromosomes represents one QTL region previously identified in mapping studies in Cvi-0 x Ler-0 RILs, with different colors representing the different mapped traits across studies. The long QTL mentioned on chromosome 5 was actually composed of smaller ones that overlap and together span the entire chromosome. We separated these and are all now visible. The *CRY2* region is indeed associated with almost all phenotypic categories.

We changed the panel (which is now Figure 6a in the new version) as suggested regarding the color of the bars and the legend and added information to aid interpretation in the caption.

Fig. 7 seems to just compile information from Fig. 6i and Fig. 2C, so it is somewhat redundant. I realize that the point of this is to show that the timing of the population expansion coincided with the allele sweeping to fixation, but it seems like this point could be made by some rearrangements of the existing figures rather than adding a new figure. There are already a lot of figures. Also, this is just an opinion, but it would make more intuitive sense to me if the figures that have time on the x-axis read time going forward from left to right, rather than back in time from left to right.

We agree that Figure 7 in the previous manuscript version was redundant, but we also find that aligning the figure panels improves clarity of the message; that population expansion coincided with the spread of the new adaptive mutation in Fogo. We therefore removed the figure from the main text and moved it to the Supplement (Supplementary Figure 14).

Concerning the direction of time on the x-axis of several figures, we agree that it is more intuitive to plot time in the forward direction, but we decided to stay with the current format, which is the standard normally used for coalescent approaches.

Page 25, lines 8-10: "...connect mutations that occurred in parallel at *FRI* and *FLC* with adaptive divergence. This case provides a symmetric example of the importance of these two genes in adaptation to aridity." There are a couple of issues here. I would say that the phenotypic shifts to earlier flowering occurred in parallel on the two islands, but the fact that there were changes in two different genes indicates that the genetic basis of the phenotypic changes is non-parallel. I think that the way that "parallel" is used here is confusing given that parallel evolution has a very

specific meaning. The genetic changes were simultaneous (roughly) but not parallel. They are also not symmetrical, and I would not call this a “symmetric example.” Instead this appears to be two separate and related but independent cases of genetic changes leading to evolutionary rescue. The phenotypic changes were the same, and the genetic pathway was even the same, but the specific genetic changes were unique to each island. I think this could be clarified and the wording improved here.

Thank you for this comment. We agree it makes sense to use a more standard approach to discuss this pattern. We made several minor changes to the text in response to this comment. These can be found on:

Page 2, line 9

Page 20, line 1 (section title)

Page 20, line 13

Page 26, line 18

Page 27, line 9

Reference 16 is missing information.

Thank you, we added the missing information

Some of the references have each word in the title capitalized, while some capitalize only the first word.

Thanks for pointing this out, we have corrected the missing information and capitalization in all references

Reviewer #3 (Remarks to the Author):

The Cvi accession of *A. thaliana* has long been known to be a special case, much diverged from other accessions. It has been frequently used in studies of functional variation. This ms now presents an interesting genetic analysis of colonization of (two) Cape Verde islands by *Arabidopsis thaliana*. The sequencing of more than 300 high quality, full genomes from the islands allows a detailed examination of the demography of colonization of first Santo Antão and then Fogo island, with estimates of N_e and timing of events. With these results available, there is an interesting genetic and functional analysis consistent with de novo mutations at two classical large-effect flowering time genes (one locus on each of the two islands colonized), rather than standing variation, having contributed to the fitness (and likely spread) of the species. The work can be of wide interest to evolutionary biologist.

We thank the reviewer for the positive feedback.

The work is put into a context of island colonization at the broadest of terms in the introduction. The ms presents a very interesting case study, in this view, $n=1$. However, for instance, Brochmann et al. (1997) write that about one half of the species on Cape Verde islands were brought by humans (331 of 621). Definitely excluding this possibility would seem to merit explicit consideration and confirmation.

Thank you for this comment. The botanical reference book by Brochmann and colleagues has been a fantastic resource that provided us with a broad understanding of the natural history of the archipelago and about many of the native and endemic species in Cape Verde. One interesting side-point here is that although herbarium samples had been collected by Wolfram Lobin and others, *A. thaliana* is not included in the set of species described in this botanical reference. This is probably because the authors did not expect that *A. thaliana* was native or endemic to the islands. Incidentally, Christian Brochmann is now a key collaborator on a project we have focused around (East African) Afroalpine populations of *A. thaliana* and in that context I had the opportunity to mention to him that his book has really been a crucial resource for us to learn about Cape Verde flora. We also currently cite this book in our manuscript.

With regard to the source of the *A. thaliana* colonists: Portuguese and Italian sailors founded Cape Verde in 1456. The Portuguese found no evidence of existing human settlements on the island when they arrived. Therefore, human introduction of *A. thaliana* is extremely unlikely based on our population genetic inferences. This is consistent with our observations that *Arabidopsis* in Cape Verde tends to occupy rocky outcrops together with island endemics and is not found in highly disturbed habitats in CVI as it is in Eurasia. The colonization timing we inferred is at least one order of magnitude older (7-10 kya), *Arabidopsis* colonization of the CV islands was therefore almost certainly a natural event, most likely mediated by wind transportation of its small seeds.

When we started the project, we expected to find recent introductions, possibly admixed with more ancient lineages. However, when we started to sequence accessions and analyze the data, we were surprised to find that the populations had such a simple phylogenetic structure without any evidence for secondary contact from the mainland. Your comments help us to realize that this is still an important point to make and to discuss that the timing does not match a human-mediated introduction.

We have added some additional discussion of the likely mode of colonization and mentioned the temporal disparity between a human-mediated origin and our findings:

Page 3, lines 18-21): We added a sentence to the introduction to describe the timing of human colonization of the islands:

The flora in CVI is a mix of native species that reached the islands via long range dispersal from mainland Africa and Macaronesia together with species introduced since 1456, when the first humans first settled in the islands (Brochmann et al., 1997).

And we added the following statement to the first section of Results:

Page 6, lines 22- page 7, lines 1-2: The inferred colonisation and dispersal timing within CVI clearly places the event well before colonization by humans, which only occurred approx. 560 years ago.

Here the data suggest that the actual origin population for *A. thaliana* is a ghost population, related to the Moroccan populations, but different. However, the model of evolution identified by their analysis identifies the existence of a ghost population as the origin of the colonizing population. How sure can we be that the mutations appeared indeed on the islands versus preexisting in the, unsampled, ghost population from mainland Africa? Could this eventuality be discussed when considering the SSWM vs WSSM alternatives?

There were similar questions from Reviewers 1 and 2. We conducted population genetic simulations using the inference of historical effective population size trajectory based on haplotype coalescences (Fig. 2c) to estimate the proportion of (selectively neutral) variants segregating in CVI that would be due to variation that already existed in the ghost population. We ran 100 whole-genome neutral coalescent simulations to estimate the number and proportion of variants segregating in the current

Santo Antão population that were retained since colonization. Simulations followed the N_e trajectory we inferred from RELATE (Fig. 2c) starting with the initial population size of 40 individuals. In our simulations, the proportion of variants segregating in present day Santo Antão population that were introduced from the ghost population was 0.017% (95% CI: 0.006% to 0.03%).

For an adaptive mutation that was present in the founding population (e.g., *FRI 232X*), the probability of fixation would be much higher and thus the probability that a selected variant that was segregating in the colonizer is still segregating today is much lower. Further, in the case of the flowering time variant, *FRI 232X*, there is additional evidence it arose recently. We do not really have space to discuss the details of within island population history in this manuscript (this will come later in other papers), but the *FRI 232X* variant is completely absent in the Cova population, which appears to have split off earliest from the other populations on the island and also appears to represent a 'relict' population within Santo Antão (Supp. Fig. 7). Therefore, it is extremely unlikely that *FRI 232X* comes from standing variation in the ghost population and difficult to invent a scenario consistent with that model. We added information about this analysis on page 6, lines 3-9

The same authors have used similar analyses earlier to examine e.g. *A. thaliana* populations from Madeira, In Madeira, the colonization seem to have been older, but there seem to have been no adaptation relating to flowering time. Some more comparison to this situation would be interesting.

Thank you for this comment. The native Madeiran accessions (there were a few recent immigrants we found) required vernalization to flower, but we do not have data that is directly comparable to the CVI or Moroccan accessions. In addition, the much higher latitude in Madeira likely results in an environment more similar to that of the relicts in Spain or North Africa. This may fit better in a future paper we are working towards where we are phenotyping accessions from mainland Eurasia and Africa, Madeira, Canary Islands and Cape Verde Islands all in the same environments.

"Intermediate" comments

-Given such a recent event, a hard sweep should be clearly observed in the two island populations. But this doesn't seem to be explored during the analyses, why?

Based on the inferred population history, including very low initial N_e and relatively rapid and recent population size change, we do not expect to have very high power to detect evidence of hard selective sweeps. Based on previous simulations, the majority of the signature from a hard selective sweep fades by $0.2 \cdot N_e$ and by $0.5 \cdot N_e$ generations there is no signal remaining in the data (see (Pennings and Hermisson 2006)).

If a sweep occurred soon after colonization in CVI, this would correspond to $0.2 \cdot 40 = 8$ to $0.5 \cdot 40 = 20$ generations in Santo Antão and $0.2 \cdot 48 = 9.6$ to $0.5 \cdot 48 = 24$ generations in Fogo. Sweeps that occurred more recently may likely be confounded with the population expansions, so may have low power for identifying sweeps with specific loci even if their trajectories were affected by positive selection. We have conducted sweep scans, but due to space constraints we did not include these results here. We plan to include these in future manuscripts focused on more recent events on CVI where sweep signatures are more relevant.

After all the filtering based on *A. lyrata*, what actual numbers of nucleotide sites are included in the analyses?

Based on the alignment of *A. lyrata* sequences to the *A. thaliana* genome, the final number of nucleotide sites used was 70676280. The number of sites lost due to missing data in the alignment of the *A. lyrata* genomes to the *A. thaliana* reference was 42064764. Such reduction in nucleotide sites is common when polarizing *A. thaliana* data to *A. lyrata*, because of the long divergence time and genomic rearrangements between the species. We also should note here that we used *A. lyrata* only in a subset of the analyses in the manuscript, where it was necessary (e.g., for site frequency spectra including Moroccan samples, both for the inference of demography and the DFE). We added information about this to the Supplement.

Suppl. fig, 12 The expression data of FLC and the phenotypes are a bit puzzling. The FRI+FLC genotype seems to have similar levels of expression as the Santa Antao samples, but what are the phenotypes of these same genotypes. How come high FLC expression consistent with early flowering (as seems to be known for Cvi-0 from earlier), a further comment would help

We have added a figure panel showing the phenotype (days to bolting) for the genotypes used in the *FLC* expression experiment (Supplementary Figure 11a) and information to the Supplementary Results pages 36-37. The Col-0 FRI+FLC+ genotype has similar *FLC* mRNA levels but different phenotype: accessions from Santo Antão carrying the FRI 232X allele flowered early despite the high levels of *FLC* observed, while FRI+FLC+ flowered late with similar *FLC* levels. Note that the late flowering observed for the S5-10 accession from Santo Antão is caused by a functional *FRI* allele.

The higher baseline levels of *FLC* in Cvi-0 have been previously described (Gazzani et al. 2003; Shindo et al. 2005) and this seems to be ancestral behavior (common also to several Moroccan accessions), see Supp Fig. 11 and page 37-38 of the Supplement.

- L5p21: "Because we take the inferred change in population size into account in our estimate of the selection coefficient, s , our estimate is conservative and likely an underestimate. For example, inference under an assumption of constant population size leads to an estimate of $s = 23.56\%$." Maybe, but don't you risk as well overestimating the selection coefficient if your inference of the intensity of the population expansion is underestimated? This second possibility would seem a bigger and more relevant concern than the (unlikely) constant population size model for these populations.

We were concerned about the case where population expansion is itself due to the variant. Now we eliminated the modelling that does not account for population growth and only mention in the text that if population growth occurred as a result of the variants, then the strength of selection would be underestimated. We also point out that this could help to explain why the selection coefficient we measure based on our fitness experiment in CVI simulated conditions suggests a much higher selection coefficient. This latter point was added to address a comment from Reviewer 4.

Minor comments

Please check references: e.g.,

The reference to the original Cvi-0 is given as Lobin (1983), rest of reference is missing.

Thank you for pointing out this mistake. We have corrected this.

- L4p27: reference in methods to the African strains is given as 8, which is a theory paper in AmNat by Uecker et al.

Thank you for pointing this out. We have corrected this.

- dN / dS ratio within pop: some criticism against this practice see (Kryazhimskiy & Plotkin 2008)

This point was also raised by Reviewer 1 and is also discussed in more detail in the introductory overview on page 3-4 of this document.

Briefly, since we are comparing each population to its phylogenetically distinct ancestor, the issues raised by Kryazhimskiy and Plotkin, who looked at the case where pairwise comparisons are made between *individuals* of the *same* phylogenetic species, are not relevant here. The most relevant points to clarify here are (1) that we compare the continental *A. thaliana* population to the *A. lyrata* outgroup, so that there is no comparison within species and (2) that *Arabidopsis* from each CVI island actually represents a distinct species from the phylogenetic perspective, i.e., the level of shared variation between each CVI population and Morocco and between the two islands is less than that observed for other species pairs in the *Arabidopsis* genus.

However, since this issue was raised by two different reviewers, we thought it was really important to ensure clarity for the expected readers of the manuscript. Therefore, we simplified the results presented and provided more rationale and explicit explanation of how the results differ from the model discussed in Kryazhimskiy and Plotkin. We also made clearer statements about the phylogenetic separation of the CVI islands from Morocco and from each other, which is important for readers to understand in the context of our dN/dS analysis.

- dN / dS: why not use a notation that reflects clearly you are using 4-fold and 0-fold degenerate positions?

We clarified that we excluded 2- and 3-fold degenerate sites from the analysis due to the uncertainty in expected counts that is introduced from asymmetries in substitution rates for this class of sites (discussed in the MEGA4 manual (<https://www.megasoftware.net/mega4/mega4.pdf>) on pages 130-136). We changed the notation for the statistic to $d_{\text{sel}}/d_{\text{neu}}$, which does not imply that we included all NS sites in the analysis and we explained more precisely what we did and why in the main text (page 13) and supplement (pages 12 and 32).

- Fig1: “n=20 per cluster”, perhaps a total number of samples used in this analysis would be nice to mention.

We added this information to the figure legend.

- Also Fig1: For the sake of helping readers understand better your results on Fig1b, why not display the individual clusters in the Eurasian group in different colors? Especially the Ir and Inr would be an interesting group to outline. This would help readers compare this tree with previous results eg Fulgione et al. 2018

Thank you for this comment. We have now plotted the tree with different colors for the INR and IR but we left other Eurasian accession the same to avoid making the plot overly complex.

Supp Fig. 13 perhaps you could consider the color scheme, can be a bit confusing at first (comment by embarrassed reviewer)

Thank you, we see now that the color scheme may have been confusing here since orange and blue were used to denote different things. We have now changed the color scheme of the tree and updated the legend to explain the new scheme.

- p3 L5: "ameliorate risk"  ameliorate risk prediction

We changed the text to: 'ameliorate predicted risks'

- p12 L7: could it be useful to clarify the origin of Ler-0?

We revised the text to clarify that *Ler-0* is a representative of the European population. The *Ler-0* accession appears to be derived from Poland, but this is not completely clear in the literature so we did not give specific details about the region of origin within Europe (page 6, lines 3-7).

The sentence now reads:

This data set allowed us to ask whether genetic polymorphisms that underlie the observed trait divergence between *Cvi-0* and other worldwide lines (with *Ler-0* as the European representative) were present in the colonizing population or whether they represent variation that arose from *de novo* mutations after colonization.

(Page 17, lines 3-7)

- L6p26: *A. thaliana* not in italic

Thank you. We have now corrected this.

Reviewer #4 (Remarks to the Author):

The authors study adaptation to dry climate in *A. thaliana* on the Cape Verde Islands. They show that two different loss of function mutations on two different islands (Santo Antao and Fogo) both lead to earlier flowering time by allowing flowering without a previous cold period. Evidence suggests that both mutations appeared *de novo* in populations of small effective size, corresponding to a strong-selection-weak-mutation regime.

This is an impressive study telling a remarkably complete story drawing on a wide range of evidence and written in a very clear style. Unfortunately, I am not familiar with the details of the molecular and computational tools employed, so I cannot really comment on the methods. I will therefore focus on the proposed evolutionary scenario.

We thank the reviewer for these positive comments.

The authors argue that the S. Antao population was founded between 7-5000 years ago from an unknown "ghost" population which had split some 40000 years ago from the Moroccan lineage. About 2000 years later (3-5 thousand years ago), the S. Antao population gave rise to the Fogo population. Then both populations acquired [sic] independent adaptations. On S. Antao a mutation of the *FRI* gene, that advances flowering time by about 35 days, appeared and rose to high frequency

in the majority of local populations. Apparently, this mutation arose after the colonization of Fogo. On Fogo, another mutation in the FLC, which advances flowering time by 27 days, arose "very soon" after the establishment of the population and subsequently went to fixation. At about the same time, the Fogo population started to expand from an initial, 1000 year long bottleneck.

Estimates of the effective population size and the mutational target size suggest that the mean waiting time for the appearance of the FLC mutation is around 3000 generations (somewhat less for the FRI mutation). Estimated selection coefficients are between 9 and 23 percent for the FRI mutation and around 4 percent for the FLC mutation. Unless selfing rate is extremely high, fixation probabilities are of the same order as the selection coefficients, and fixation times are short (around 50 generations). So, while the mean waiting time for a successful mutation might well be on the order of 10000 generations or higher, such mutations apparently appeared considerably faster in the Cape Verde populations, and these fortuitous events might have contributed to long-term population survival.

Thank you for the summary. There are a few points where some details don't quite match with our inferences or are switched around (e.g., the inferred selection coefficients of the two variants are switched and the 10kya wait time is not something we find) but overall this summary is broadly consistent with what we show. Based on some confusion here we thought that we needed to provide a little bit more detail about the population history findings and revised the text on page 6 for this reason (changes marked in red).

As for the colonization and appearance of adaptive mutations, as the reviewer notes, 'fortuitous' events are indeed important here! For example, there may have been many (failed) dispersal events to Cape Verde but only a small fraction of those could have led to a viable population based on the proportion of the landmass that could support an *A. thaliana* population, especially one that was not locally adapted. Similarly, mutations that result in major reductions in flowering time are chance events with low probability in any given individual and generation. As discussed in (Gomulkiewicz and Holt 1995), local extinction events after colonization are incredibly difficult to study, especially those that occur just after dispersal, even though these are likely common. In nature, we can only observe those events that result in viable populations (i.e., those that escaped extinction). We hope that the findings will be useful as a starting point to collect information about diverse cases of long-range colonization of novel habitats and we appreciate the reviewers' supportive comments about these aspects of the project. Related to these comments, we now mention the importance of chance events in adaptation (page 25, lines 7-8).

I don't have any serious criticisms of this scenario, just a few questions/comments/clarifications:

- Given that both populations survived for about 1000 generation before the appearance of the two mutations, I find the term "evolutionary rescue" a bit strained.

We appreciate this comment but it seems to be a point that was explicitly supported by other reviewers. Also, our new climate analyses lend further support to the idea that a major reduction in flowering time would have been required for *Arabidopsis* to establish stable populations on CVI.

Based on our reconstructions and inference, both mutations appeared when the island populations were stagnating at N_e of approx. 40, which would have meant they were unstable and susceptible to extinction. The reconstructed N_e trajectories only increase at around the time that the *FRI* and *FLC* variants appear. Overall, based on the combination of (new) climate analyses, population genetic results and fitness differentials for these variants we believe that there is strong evidence that the variants were important for establishing long-term stable populations on the islands.

- I might have missed it, but are there any data about the phenotype and relative fitness of *S. Antao* individuals that do not carry the *FRI* mutation?

This can be found in Fig. 7e and the associated source data file. They flower later and have lower fitness – on average 34 days later and 387 fewer seeds – which is similar to the difference in Col-0 (*FRI*-) and a Col-0 where a functional *FRI* allele is introduced (27 days later and 669 fewer seeds).

- Can one say something about the ratio of effective to census population size?

We revisited the same sites over the course of 7 years (2012-2019). Although the sub-populations at each site were very stable over the full term of the study, there was a lot of variation from year-to-year in the density of plants at a given site and in some especially dry years *A. thaliana* was completely absent across most sites. We do not have systematic data to estimate the census population size and any estimate would need to somehow deal with what might be happening on some of the inaccessible cliff faces and steep ravines where *Arabidopsis* is likely present. In a very good year, we roughly estimated there must be approx. 100k plants setting seed. In other years the number of plants in the field could conceivably be on the order of our population genetic estimates of N_e , i.e., approx. 5000 per island. Since we do not have a robust estimate of these numbers and there are a lot of assumptions embedded in them about the limits of the distribution and inaccessible locations, we don't really feel comfortable mentioning this in the manuscript text.

- page 10, 10: How can the results from the fitness assays (8- and 16-fold advantage of CVI forms) be reconciled with estimates of selection coefficients ("only" 0.09-0.23)? See also 140 increase in seed number for the *FRI* mutation (page 19, 10).

One possibility is that the variants themselves caused population expansion (consistent with an evolutionary rescue scenario) so that we underestimate the selection coefficients when we use a population genetic approach that controls for overall population expansion. Another factor that may contribute is environmental heterogeneity across space and time and the possibility that a seed bank could allow some genotypes to successfully reproduce (i.e., to germinate, flower and set seed) only in moister years. However, for our fitness experiment, we simulated one of the moistest years, when plants at the site and across the islands overall were able to set seed.

We now mention the issue of inferring s using an approach that controls for overall population expansion in the case where the variants *cause* the population expansion in the Results (page 22, line 12-19) and have discussed some of the other possibilities in the Supplementary Information.

Minor comments:

- 9, 18: What do the negative percentages in parenthesis mean?

This question refers to the negative percentage of adaptive substitutions that we had reported. As the reviewer correctly points out, negative percentages are not logical here. This was the raw output of polyDFE but should be interpreted as 0; negative values are due to noise around the estimate. Updated text can be found on page 14, line 9.

- 9, 19: DFE of what? segregating variants?

Yes, the DFE is based on the site frequency spectrum, which is calculated from segregating variants. Changes are marked in red on page 14, line 10.

- 10, 13: say something about how much higher fitness of local plants was in Morocco (the difference is much weaker than on CVI)

Compared to the fitness differential in CVI conditions, the differential was not minor and not significant in Moroccan conditions. However, Moroccan plants did perform somewhat better than CVI plants in Moroccan conditions. Mean fitness was 1.5-fold higher for Moroccan plants in Moroccan conditions than for Santo Antão representatives and 1.7-fold higher than Fogo plants. This is now shown in Figure 5d.

- 12, 22: enrichment with respect to what?

We now clarify this in the text:

Page 17, lines 19-21: 'At QTL mapping intervals, which cover most of the genome, we found very slight and non-significant enrichment of private variation relative to the genome-wide proportion...'

- 24, 18: what do you mean by symmetric?

We have changed the text to refer to genetic convergence, which is a more standard way to describe this (page 27, line 9).

- 29, 15: something seems to be missing before "and the R package ape"

We have edited this sentence. Since all neighbor-joining trees were produced using ape, we simplified the sentence as follows:

Page 32, lines 12-13: 'We produced neighbor-joining trees using the R package ape v.3.5 (Paradis, Claude, and Strimmer 2004) (<https://github.com/HancockLab/CVI>).'

- 31, 5: I'm a bit confused by the "upper" and "lower" bound to the colonization time. Which one is closer to the present?

We apologize for the confusion and we clarified this in the manuscript. We were thinking in terms of the tree structure and therefore used "upper" for "more ancient", and "lower" for "more recent" bounds. We added mention of this to the main text, page 5, lines 13 and 20-21.

References

- 1001 Genomes Consortium. 2016. “1,135 Genomes Reveal the Global Pattern of Polymorphism in *Arabidopsis Thaliana*.” *Cell* 166 (2): 481–91. <https://doi.org/10.1016/j.cell.2016.05.063>.
- Brennan, Adrian C., Belén Méndez-Vigo, Abdelmajid Haddioui, José M. Martínez-Zapater, F. Xavier Picó, and Carlos Alonso-Blanco. 2014. “The Genetic Structure of *Arabidopsis Thaliana* in the South-Western Mediterranean Range Reveals a Shared History between North Africa and Southern Europe.” *BMC Plant Biology* 14 (January): 17. <https://doi.org/10.1186/1471-2229-14-17>.
- Dessandier, P.-A., J. Knies, A. Plaza-Faverola, C. Labrousse, M. Renoult, and G. Panieri. 2021. “Ice-Sheet Melt Drove Methane Emissions in the Arctic during the Last Two Interglacials.” *Geology* 49 (7): 799–803. <https://doi.org/10.1130/G48580.1>.
- Durvasula, Arun, Andrea Fulgione, Rafal M. Gutaker, Selen Irez Alacakaptan, Pádraic J. Flood, Célia Neto, Takashi Tsuchimatsu, et al. 2017. “African Genomes Illuminate the Early History and Transition to Selfing in *Arabidopsis Thaliana*.” *Proceedings of the National Academy of Sciences of the United States of America* 114 (20): 5213–18. <https://doi.org/10.1073/pnas.1616736114>.
- Eyre-Walker, Adam. 2002. “Changing Effective Population Size and the McDonald-Kreitman Test.” *Genetics* 162 (4): 2017–24.
- Gazzani, Silvia, Anthony R. Gendall, Clare Lister, and Caroline Dean. 2003. “Analysis of the Molecular Basis of Flowering Time Variation in *Arabidopsis* Accessions.” *Plant Physiology* 132 (2): 1107–14. <https://doi.org/10.1104/pp.103.021212>.
- Gomulkiewicz, Richard, and Robert D. Holt. 1995. “When Does Evolution by Natural Selection Prevent Extinction?” *Evolution* 49 (1): 201. <https://doi.org/10.2307/2410305>.
- Hancock, Angela M., Benjamin Brachi, Nathalie Faure, Matthew W. Horton, Lucien B. Jarymowycz, F. Gianluca Sperone, Chris Toomajian, Fabrice Roux, and Joy Bergelson. 2011. “Adaptation to Climate across the *Arabidopsis Thaliana* Genome.” *Science (New York, N. Y.)* 334 (6052): 83–86. <https://doi.org/10.1126/science.1209244>.
- Intergovernmental Science-Policy Platform on Biodiversity and Ecosystem Services, IPBES. 2019. “Summary for Policymakers of the Global Assessment Report on Biodiversity and Ecosystem Services.” Zenodo. <https://doi.org/10.5281/ZENODO.3553579>.
- Konapala, Goutam, Ashok K. Mishra, Yoshihide Wada, and Michael E. Mann. 2020. “Climate Change Will Affect Global Water Availability through Compounding Changes in Seasonal Precipitation and Evaporation.” *Nature Communications* 11 (1): 3044. <https://doi.org/10.1038/s41467-020-16757-w>.
- Novikova, Polina Yu, Nora Hohmann, Viktoria Nizhynska, Takashi Tsuchimatsu, Jamshaid Ali, Graham Muir, Alessia Guggisberg, et al. 2016. “Sequencing of the Genus *Arabidopsis* Identifies a Complex History of Nonbifurcating Speciation and Abundant Trans-Specific Polymorphism.” *Nature Genetics* 48 (9): 1077–82. <https://doi.org/10.1038/ng.3617>.
- O’Neill, Brian C., Claudia Tebaldi, Detlef P. van Vuuren, Veronika Eyring, Pierre Friedlingstein, George Hurtt, Reto Knutti, et al. 2016. “The Scenario Model Intercomparison Project (ScenarioMIP) for CMIP6.” *Geoscientific Model Development* 9 (9): 3461–82. <https://doi.org/10.5194/gmd-9-3461-2016>.
- Paradis, Emmanuel, Julien Claude, and Korbinian Strimmer. 2004. “APE: Analyses of Phylogenetics and Evolution in R Language.” *Bioinformatics* 20 (2): 289–90. <https://doi.org/10.1093/bioinformatics/btg412>.
- Pennings, Pleuni S., and Joachim Hermisson. 2006. “Soft Sweeps III: The Signature of Positive Selection from Recurrent Mutation.” *PLoS Genetics* 2 (12): e186. <https://doi.org/10.1371/journal.pgen.0020186>.

- Rousselle, Marjolaine, Maeva Mollion, Benoit Nabholz, Thomas Bataillon, and Nicolas Galtier. 2018. "Overestimation of the Adaptive Substitution Rate in Fluctuating Populations." *Biology Letters* 14 (5): 20180055. <https://doi.org/10.1098/rsbl.2018.0055>.
- Schiermeier, Quirin. 2021. "Climate Change Made North America's Deadly Heatwave 150 Times More Likely." *Nature*, July, d41586-021-01869-0. <https://doi.org/10.1038/d41586-021-01869-0>.
- Schuur, E. A. G., A. D. McGuire, C. Schädel, G. Grosse, J. W. Harden, D. J. Hayes, G. Hugelius, et al. 2015. "Climate Change and the Permafrost Carbon Feedback." *Nature* 520 (7546): 171–79. <https://doi.org/10.1038/nature14338>.
- Shindo, Chikako, Maria Jose Aranzana, Clare Lister, Catherine Baxter, Colin Nicholls, Magnus Nordborg, and Caroline Dean. 2005. "Role of *FRIGIDA* and *FLOWERING LOCUS C* in Determining Variation in Flowering Time of Arabidopsis." *Plant Physiology* 138 (2): 1163–73. <https://doi.org/10.1104/pp.105.061309>.
- Sperle, Thomas, and Helge Bruelheide. 2021. "Climate Change Aggravates Bog Species Extinctions in the Black Forest (Germany)." Edited by Amanda Taylor. *Diversity and Distributions* 27 (2): 282–95. <https://doi.org/10.1111/ddi.13184>.
- Steinbach, Julia, Henry Holmstrand, Kseniia Shcherbakova, Denis Kosmach, Volker Brüchert, Natalia Shakhova, Anatoly Salyuk, et al. 2021. "Source Apportionment of Methane Escaping the Subsea Permafrost System in the Outer Eurasian Arctic Shelf." *Proceedings of the National Academy of Sciences* 118 (10): e2019672118. <https://doi.org/10.1073/pnas.2019672118>.
- Tabari, Hossein. 2020. "Climate Change Impact on Flood and Extreme Precipitation Increases with Water Availability." *Scientific Reports* 10 (1): 13768. <https://doi.org/10.1038/s41598-020-70816-2>.
- Tollefson, Jeff. 2020. "How Hot Will Earth Get by 2100?" *Nature* 580 (7804): 443–45. <https://doi.org/10.1038/d41586-020-01125-x>.
- Urban, M. C. 2015. "Accelerating Extinction Risk from Climate Change." *Science* 348 (6234): 571–73. <https://doi.org/10.1126/science.aaa4984>.
- Walter, K. M., S. A. Zimov, J. P. Chanton, D. Verbyla, and F. S. Chapin. 2006. "Methane Bubbling from Siberian Thaw Lakes as a Positive Feedback to Climate Warming." *Nature* 443 (7107): 71–75. <https://doi.org/10.1038/nature05040>.

Reviewers' Comments:

Reviewer #1:

Remarks to the Author:

The revised manuscript is very well presented and tells a coherent and fascinating story of adaptation. I would like to thank the authors for their expert analyses, the inclusion of new results, and their thorough revision of the text which is now much clearer and easier to follow.

I have just two minor comments:

I was surprised to read in the first sentence of the introduction that natural environmental change is damaging to biodiversity. ("One in eight of the world's existing plant and animal species are at risk of extinction due to natural and human-mediated environmental change¹."). I have checked the report and could not find this point, but may have missed it. Please clarify. Also, the suggested citation for this report according to <https://ipbes.net/global-assessment> is 2019, not 2020.

Legend to Figure 4c: I suggest something like "Predicted climate suitability for Moroccan Arabidopsis in CVI ..."

Reviewer #2:

Remarks to the Author:

The authors have substantially revised and improved this manuscript. They added helpful analyses such as niche modeling as well as coalescent simulations to estimate the probability that segregating variation derived from existing variation versus new mutations. They clarified a number of key points and revised the text and figures. They gave detailed responses to reviewer comments and addressed all reviewer concerns. I think this is an impressive and interesting study that documents and provides details of two related cases in which rapid evolution appears to have rescued populations, which is a novel and important contribution to the field.

Reviewer #3:

Remarks to the Author:

The authors have carefully considered the reviewer comments, broadened the perspective of the ms in some areas, modified the text, and added some analyses, and thus improved the ms in many ways. There are three bigger issues and some minor comments

1) The revision has now a suggestion that they are dealing with phylogenetic species. It seems that the issue of phylogenetic species is perhaps confusing. The major genetic change is loss of variation due to bottlenecks and drift. It seems much better justified to deal with SA and F populations as highly diverged sets of selfing lines rather than new species. If new species, then it seems other analyses would need to be modified, and the results should be discussed also in terms of speciation concepts in the case of selfing plants.

2) Perhaps this view of species status was changed to further justify the Dsel/Dneu analysis, by showing that the entities to be compared are quite diverged?

The genomewide Dsel/Dneu are exceptionally high, about 2, compared to most other species (often about 0.2-0.4), as is the proportion of loci fixed due to positive selection (70 %, earlier very low estimates for *A. thaliana*, e.g., around 0 in Bustamante et al. 2002, for *Drosophila* earlier a very high close to 50 %). Reviewer 1 already in the first round pointed to lack of alternative explanations, The high Dsel/Dneu is ascribed here mainly to positive selection (and now, an addition on the effect of deleterious allele fixation, but the small addition here does not seem to be sufficient. The estimates of Dsel/Dneu (around 2) and $\alpha = 70\%$ would require much more comparison to earlier results, evaluation of methods and analysis of caveats (review with discussion of effects of demographic

history for instance in Zhen et al and Lohmueller 2021 Genome Res).

Timing matters: The authors state (p. 14, L. 1)"However, given the phylogenetic separation between CVI populations and the Moroccan outgroup this is not relevant here.". The time since the split is also important to consider (Brandvain and Wright, 2016 Trends in Genetics) and Mugal et al. 2014, 2020, Molecular Biology and Evolution) as after a population bottleneck, Dsel reaches its equilibrium ("real") value faster than Dsyn. In the case of a selfer and after such strong bottleneck, it may be indeed that this is not a concern. But given these high values obtained, it would be relevant to discuss potential causes for biases.

The authors report these exceptional estimates with little detail on the methods and little comparison to earlier results or other discussion (aside from p. 27, lines 11-12), nothing in the abstract. Qre these analyses even necessary for the main points of the ms? It seems the main message does not need them, and they do not write much about them.

3) The climate modelling is based on current climates, as no historical data seem available. Perhaps some caveat could be included with these strong statements.

"We sequenced genomes of 335 newly collected wild Arabidopsis samples from the Cape Verde Islands to investigate the mechanisms of adaptation after a sudden shift to a more arid climate"

And p15, L11-13"Taken together these results highlight the challenging climatic conditions plants would have faced upon colonization of CVI, consistent with results from niche analysis (Fig. 4)."

minor comments

p. 14 L8-14. The DFE is described after this text, even though alpha estimates are based on the DFE. The description of the DFE could be clearer, and relate to strength of mutational effects.

p. 15, line 9 : Marocco was slightly better than CVI , but $P > 0.67$ – is it not better to say that both had equal fitness

Fig 4. The color scheme is a bit confusing: in fig 4b,c red reflects areas of high suitability for the species, while in Fig 4.d, red reflects high dissimilarity and has a very different (opposite) interpretation.

p31 L11: curiously, the reference for ape is missing (Paradis et al. 2004) (although present in the rebuttal, and present in the bibliography of the main manuscript as reference 98)

p35 L20: the reference 98 (Paradis et al. 2004, for the Ape R package) seems incorrect

Reviewer #4:

Remarks to the Author:

I am very happy with the revision and do not have much else to say. My only comment is the following: The presumed 10000 year waiting time for a successful new mutation I mentioned in my previous report came from the following reasoning. According to Supplementary Table 11 (table 35 in the previous version), the mean time to appearance of a new mutation (Mean gen. to 1st seg. variant) is around 3000 generations, and the fixation probability (%fixed) is around 20%, so I gathered that the time to first fixation would be around $3000/0.2 = 15000$ generations. I now see this is actually in contradiction with the column "Mean gen. to adapt", which is quite similar to Mean gen. to 1st seg. variant. But now I don't understand where is the error in my reasoning, and why the two times are so close.

Minor comments:

- 6, 9: in CVI
- Fig. 1: what are relics?
- 15, 4: Do you mean biotic factors? (The same formulation also appears later in the discussion.)
- 23, 4: is consistent

We thank the reviewers for their comments on the revised version of our manuscript. Overall the comments from the reviewers were very positive. There were a few requests for clarification and suggestions for further improvement. Just below is a summary of the reviewer's comments and our responses. Full reviewer comments and our complete responses can be found in the 'REVIEWER COMMENTS' section below that.

Reviewer #1 was mostly satisfied with the revised manuscript but pointed out a couple of minor issues, which we corrected based on the suggestions.

Reviewer #2 was satisfied with the revised manuscript and made several positive comments about it.

Reviewer #3 had a generally positive impression of the revised manuscript but also introduced some points related to our assertion that the patterns in the Cape Verde Arabidopsis lineages fit a 'phylogenetic species' concept. We changed this terminology to instead mention that the lineages were monophyletic, which we expect should be less contentious than opening a discussion of speciation and species concepts.

Reviewer #3 also suggested we add a caveat about the climatic differences between Morocco and CVI related to potential difference between present and past climate and we have added a statement about this to the SDM section.

Reviewer #4 was generally positive but noted some confusion about some of the results we reported in Table 11. We re-examined this table and the annotation of the column headers and realized this was unclear. We have now revised the summary information for the table and address the specific question from the reviewer below.

All changes are marked in red in the revised version of the manuscript.

REVIEWER COMMENTS

Reviewer #1 (Remarks to the Author):

The revised manuscript is very well presented and tells a coherent and fascinating story of adaptation. I would like to thank the authors for their expert analyses, the inclusion of new results, and their thorough revision of the text which is now much clearer and easier to follow.

Thank you. We agree that the suggestions from the reviewers improved the manuscript, and we appreciate your suggestions from the previous round!

I have just two minor comments:

I was surprised to read in the first sentence of the introduction that natural environmental change is damaging to biodiversity. ("One in eight of the world's existing plant and animal species are at risk of extinction due to natural and human-mediated environmental change¹."). I have checked the report

and could not find this point, but may have missed it. Please clarify. Also, the suggested citation for this report according to <https://ipbes.net/global-assessment> is 2019, not 2020.

Thank you. It seems that we were aiming to be conservative and not overstate the impact of human-mediated change but in doing so made an incorrect statement. We have changed this to read:

One in eight of the world's existing plant and animal species are at risk of extinction due to human-mediated environmental change¹.

It is also true that the date for this reference was incorrect. We initially included this in the manuscript based on notes from the oral presentation of the report at the IPBES Plenary in May 2019 (IPBES-7). This was before the final version of the report was released and we were not careful enough in updating the information. We have corrected this date.

Legend to Figure 4c: I suggest something like "Predicted climate suitability for Moroccan Arabidopsis in CVI ..."

Thank you for the suggestion. We changed this in the revised manuscript.

Reviewer #2 (Remarks to the Author):

The authors have substantially revised and improved this manuscript. They added helpful analyses such as niche modeling as well as coalescent simulations to estimate the probability that segregating variation derived from existing variation versus new mutations. They clarified a number of key points and revised the text and figures. They gave detailed responses to reviewer comments and addressed all reviewer concerns. I think this is an impressive and interesting study that documents and provides details of two related cases in which rapid evolution appears to have rescued populations, which is a novel and important contribution to the field.

Thank you. And thank you for your helpful comments in the first round.

Reviewer #3 (Remarks to the Author):

The authors have carefully considered the reviewer comments, broadened the perspective of the ms in some areas, modified the text, and added some analyses, and thus improved the ms in many ways.

Thank you.

There are three bigger issues and some minor comments

1) The revision has now a suggestion that they are dealing with phylogenetic species. It seems that the issue of phylogenetic species is perhaps confusing. The major genetic change is loss of variation due to bottlenecks and drift. It seems much better justified to deal with SA and F populations as highly diverged sets of selfing lines rather than new species. If new species, then it seems other analyses would need to be modified, and the results should be discussed also in terms of speciation concepts in the case of selfing plants.

Thank you for these comments. We are glad to have the opportunity to clarify these points and to avoid confusion.

The 'phylogenetic species concept' is really just a reference to the case where lineages are monophyletic, but there is a good deal of controversy around species concepts and these are not central to our point. Therefore, we have changed this sentence to be more descriptive/functional. Now we simply point out that each CVI island lineage forms a highly diverged, monophyletic lineage and is thus phylogenetically distinct. Thus, we avoid any mention of species concepts here but hopefully convey the message that the CVI lineages fit with the assumptions of phylogenetic approaches like dN/dS or d_{sel}/d_{neu} . That is, they almost completely lack shared segregating polymorphism.

We do not delve into the question of whether these lineages would represent species, but it is also the case that most defined endemic species in Cape Verde would likely not represent species under a strict biological species concept definition, because they are inter-compatible. This is not so surprising because plants are known for maintaining compatibility even across long evolutionary time-scales.

The reviewer refers to the lineages as 'selfing lines', which seems to suggest that there is no variation within each island population – but maybe this was unintentional. In any case, we will address this so that it can be resolved in case it was seen as an issue. While it is true that variation in CVI *Arabidopsis* lineages is low, and that the patterns are in some ways simpler than in continental populations (e.g., no introgression from other populations), we actually think that a really neat aspect of these island populations is that their rather recent origin and near complete colonization bottlenecks allow us to reconstruct how variation built up over time. These aspects will become clearer in future manuscripts involving these populations. Regarding the concern about self-compatibility in the *Arabidopsis* lineages and whether this makes them somehow different from other colonizing lineages, we would like to mention that it is known that self-compatibility is over-represented among island colonists/endemic species. This is known as 'Baker's Law', which states that species that colonize islands by long-distance dispersal are likely to be self-compatible due to the advantage in reproductive assurance. In other words, obligate outcrossers are unlikely to be successful long-range colonizers (in the initial stages of colonization) because they would be unable to reproduce in the new environment (Baker, *Evolution*, 1955, doi: 10.2307/2405656). Therefore, the fact that *Arabidopsis* is self-compatible does not make it distinct from other natural examples of island radiations and speciation processes.

We also thought it would be helpful for the readers if we briefly pointed out that the pattern we observe for *Arabidopsis* in CVI is analogous to the patterns in plant lineages that are classified as endemic species in Cape Verde and in island systems in general. Most Cape Verdean endemics are classified as 'schizoendemic species', which are defined as species that formed from a more widely distributed taxon after reproductive isolation due to physical separation. It has been estimated that 90.4% of the endemic species in Cape Verde are likely to be schizoendemics (Brochmann et al., 1997, *Sommerfeltia*, page 66). *Arabidopsis* lineages, similar to most formally recognized schizoendemics in CVI show clear eco-distributional patterns (Brochmann 1997, *Sommerfeltia*, page 66).

In case this would seem like a step too far, I contacted Christian Brochmann to get his opinion about this idea and the specific wording of this statement. He agreed that the pattern we observe for *Arabidopsis* in Cape Verde is similar to the common schizoendemic pattern observed for other species and he suggested a couple of additional relevant references, which we added to the text in addition to his 1997 botanical reference. We also added Dr. Brochmann to the Acknowledgements for helpful discussions on this point.

2) Perhaps this view of species status was changed to further justify the Dsel/Dneu analysis, by showing that the entities to be compared are quite diverged?

As we explained in the previous responses and above here, we realized from the previous round of reviews that it was difficult for reviewers to understand the implications for the patterns of differentiation we observe between the two island lineages. More generally, we were looking for terminology that fits this case and also helps readers understand the reasoning behind the choice of statistical approaches. For example, in the past, when we simply referred the two island lineages as 'populations' in seminar or conference presentations, we have often gotten questions about why we should conduct GWAS separately between islands.

We therefore hoped that pointing out that these lineages are monophyletic and fit with a phylogenetic species concept could help the reader to intuitively understand the rationale behind our analysis approaches and how to interpret our results. For example, all of these issues, including the ones raised here and in the previous round related to dN/dS -type statistics, seem to result from a lack of clarity on our part that these lineages are monophyletic, which is an important assumption of this family of approaches. However, due to the reviewer's concerns, in the revised manuscript we have opted for a less controversial phrasing to describe this. We now simply refer to the lineages as 'monophyletic' and 'phylogenetically distinct' without any reference to species terminology.

The genomewide Dsel/Dneu are exceptionally high, about 2, compared to most other species (often about 0.2-0.4), as is the proportion of loci fixed due to positive selection (70 %, earlier very low estimates for *A. thaliana*, e.g., around 0 in Bustamante et al. 2002, for *Drosophila* earlier a very high close to 50 %). Reviewer 1 already in the first round pointed to lack of alternative explanations, The high Dsel/Dneu is ascribed here mainly to positive selection (and now, an addition on the effect of deleterious allele fixation, but the small addition here does not seem to be sufficient. The estimates of Dsel/Dneu (around 2) and $\alpha = 70\%$ would require much more comparison to earlier results, evaluation of methods and analysis of caveats (review with discussion of effects of demographic history for instance in Zhen et al and Lohmueller 2021 Genome Res).

The discrepancies between our results and previous results for *A. thaliana* samples taken from across its species range or even the more limited Eurasian continental range seem to us reasonable given that gene flow across the landscape is low and the continental range is large, so that mutations that arise are unlikely to spread across the species distribution of *A. thaliana* and fix even if they are adaptive. This pattern is common in plant species where most migration occurs only over short-distances by seeds or pollinators (but not wind-dispersed pollen). The history of *Arabidopsis* lineages in Cape Verde is very different from that of continental *A. thaliana*. Here, we have a case where there was a strong bottleneck combined with what seems to be a large change in climate compared to the

progenitor. The combination of these two factors might be expected to result in either extinction of the colonizing lineage or adaptation. In the case of adaptation, the bottleneck(s) and initial small size of the colonizing population(s) would have produced exactly the conditions necessary for dN/dS to be high.

This paper is already well over the length normally requested by *Nature Communications* due to the many additions we made to address the concerns in the first round of reviews. We are concerned about space and that detailed discussions of comparisons to previous work in other systems would fit better in a future review/synthesis paper. However, we agree with the reviewer that it is prudent to be conservative and so we added a sentence at the end of the section that points out that, while the estimates are consistent with strong positive selection, the confidence intervals are large as a result of the limited numbers of fixed and segregating sites in the CVI lineages overall as exhibited in Fig. 5b-c.

Timing matters: The authors state (p. 14, L. 1) "However, given the phylogenetic separation between CVI populations and the Moroccan outgroup this is not relevant here."

This statement in the paper refers specifically to the issues raised in the Plotkin paper, which are unrelated to timing.

The time since the split is also important to consider (Brandvain and Wright, 2016 Trends in Genetics) and Mugal et al. 2014, 2020, Molecular Biology and Evolution) as after a population bottleneck, D_{sel} reaches its equilibrium ("real") value faster than D_{syn} . In the case of a selfer and after such strong bottleneck, it may be indeed that this is not a concern. But given these high values obtained, it would be relevant to discuss potential causes for biases.

The issue raised by Mugal et al., 2014 and Brandvain and Wright 2016 relates to the potential problems that arise after an *incomplete* bottleneck. In this case, as the reviewer states, this is not an issue because the bottleneck was 99% complete, so that almost no polymorphisms are segregating between the mainland and island populations. Unfortunately, we are well over the suggested word limit for *Nature Communications* and it seems that, while this is an interesting topic, as the reviewer notes, it is unlikely to be relevant in this specific case.

The authors report these exceptional estimates with little detail on the methods and little comparison to earlier results or other discussion (aside from p. 27, lines 11-12), nothing in the abstract. Are these analyses even necessary for the main points of the ms? It seems the main message does not need them, and they do not write much about them.

The methods are briefly described on page 35, lines 4-11 and additional details are provided in the Supplementary Text on pages 12-13. In addition to the results on page 27 of the main text, we discuss the results in the Supplementary Text on pages 32-34.

We feel that the evidence of adaptive evolution on the branches leading to the CVI islands is a central point and that it is consistent with other aspects of the data (i.e., climatic divergence of CVI compared to Morocco, signal from fitness experiments, signal from the QTL analysis). We also

believe it is an important component for understanding the overall picture. There is a tight length constraint for the abstract so we are necessarily brief, but the statement:

Patterns of genetic variation within and between islands revealed a polygenic response to multivariate selection.

in the abstract is meant to refer both to the result from d_{sel}/d_{neu} and the QTL analysis because both show a signature of adaptation. We also discuss this point in the 4th paragraph of the Discussion where we refer to Figs. 5a-b and 6c.

Finally, these results are important because they help to set up the background for future papers from our lab and other labs that will use the associated genetic resources. We have been working toward this manuscript for nearly 10 years – the idea came from a 2011 paper about climate adaptation where I found that Cvi-0 was an extreme outlier – and we have spent quite a lot of time working to make this paper as useful and comprehensive as possible as an introduction to these populations. We believe that including the results in this section is important as part of the goal to provide the whole story and a strong foundation for future work on the CVI lineages.

3) The climate modelling is based on current climates, as no historical data seem available. Perhaps some caveat could be included with these strong statements.

"We sequenced genomes of 335 newly collected wild Arabidopsis samples from the Cape Verde Islands to investigate the mechanisms of adaptation after a sudden shift to a more arid climate" And p15, L11-13"Taken together these results highlight the challenging climatic conditions plants would have faced upon colonization of CVI, consistent with results from niche analysis (Fig. 4)."

We added a caveat to the SDM section regarding the lack of specific and reliable historical information about these environmental variables. The addition is pasted below and can be found in the manuscript on page 11:

Although there is the possibility that at the time of colonization the climates were somewhat more similar, or that the Moroccan population extended into more extreme climatic zones, based on our results using present-day data, there are large differences in many aspects of climate in CVI relative to Morocco.

minor comments

p. 14 L8-14. The DFE is described after this text, even though alpha estimates are based on the DFE. The description of the DFE could be clearer, and relate to strength of mutational effects.

We have changed the order where we discuss results on the DFE and alpha following the reviewer suggestions. We agree with the reviewer that this ordering is more logical.

p. 15, line 9 : Morocco was slightly better than CVI , but $P > 0.67$ – is it not better to say that both had equal fitness

Thank you for the suggestion. We changed this in the manuscript to read:

There was no significant difference in fitness for the Moroccan and CVI lines in the Moroccan simulated environment ($\beta=0.337$, $P=0.679$).

Fig 4. The color scheme is a bit confusing: in fig 4b,c red reflects areas of high suitability for the species, while in Fig 4.d, red reflects high dissimilarity and has a very different (opposite) interpretation.

We were also unsure about this in the beginning (before the previous resubmission) but decided on this color scheme because it is the standard one used for plotting these statistics and also seemed to be clearer than some others we tried.

p31 L11: curiously, the reference for ape is missing (Paradis et al. 2004) (although present in the rebuttal, and present in the bibliography of the main manuscript as reference 98)

p35 L20: the reference 98 (Paradis et al. 2004, for the Ape R package) seems incorrect

We are not sure how this happened but appreciate that you found these issues and pointed them out. We have corrected the problems in the revised version.

Reviewer #4 (Remarks to the Author):

I am very happy with the revision and do not have much else to say.

Thank you!

My only comment is the following: The presumed 10000 year waiting time for a successful new mutation I mentioned in my previous report came from the following reasoning. According to Supplementary Table 11 (table 35 in the previous version), the mean time to appearance of a new mutation (Mean gen. to 1st seg. variant) is around 3000 generations, and the fixation probability (%fixed) is around 20%, so I gathered that the time to first fixation would be around $3000/0.2 = 15000$ generations. I now see this is actually in contradiction with the column "Mean gen. to adapt", which is quite similar to Mean gen. to 1st seg. variant. But now I don't understand where is the error in my reasoning, and why the two times are so close.

The values in the %fixed columns refer to the percent of all simulations in which an adaptive variant both arose and fixed rather than the percent of times the adaptive variant fixed once it arose. We have changed the annotation in Supp. Table 11 as follows to better explain this.

Changes to definition of %fixed in Supp Table 11:

previous: % fixed: the percentage of simulations where the adaptive variant fixed

current version: % fixed: the percentage of simulations where an adaptive variant arose and fixed in the population

Minor comments:

- 6, 9: in CVI

Added

- Fig. 1: what are relicts?

We changed the text to clarify this. It now reads:

Divergent Iberian lines, or relicts, are shown in dark green, other Iberian lines (non-relicts) are shown in purple, and all other Eurasian lines in magenta.

- 15, 4: Do you mean biotic factors? (The same formulation also appears later in the discussion.)

Yes, thank you!

- 23, 4: is consistent

Changed, thank you for pointing this out.

Reviewers' Comments:

Reviewer #3:

Remarks to the Author:

Thanks for the extensive comments on parts of the earlier reviewer comments.

There was an issue about taking up phylogenetic species, mainly because this seemed to bring up issues outside the main focus of the ms. Describing SA and FO as highly diverged populations seems like an appropriate way to deal with the issue.

Questions about dN/dS , DFE and α were brought up in the first round, more extensively in the second. In the revision, the authors have added one sentence, but mainly have not discussed the exceptional findings, or to replied to the comments on the difficulty of interpretation.

To repeat, the different methods for estimating DFE and α have several assumptions. Here they used polyDFE, which, as others, also assumes a set of independent SNPs. Thus, it is assumed that some sites are neutral, while others undergo independently negative selection, others positive selection, as reflected in the DFE. The model allows the expected neutral and deleterious substitutions (due to drift), and then the rest are assumed to be due to positive selection. This holds, given independence (lack of high linkage disequilibrium between loci) (and some other assumptions). (of course, in most cases these do not hold exactly).

Collections of selfing species over wide distances have moderate LD (e.g. Arabidopsis, Nordborg et al. 2002 Nature Genetics, Medicago, Branca et al. 2011 PNAS, because of considerable historical recombination). Small local populations of *A. thaliana* have high LD, which means that the sites are not independent (e.g. Nordborg et al 2002 Nature Genetics, Bomblies et al. 2010 PLoS Genet). When there is much rapid selection over some area in the genome (as the flowering time loci here), the effects are expected to spread over large areas of the genome when effective recombination is low and LD is high, as expected to be here. So there is opportunity for strong selection, as the authors accurately state, but also large opportunity for effects due to nonindependence of loci (data are not shown). (And of course, for other demographic effects). The shape of the DFE and the surprisingly high estimates of α (60-50 %) are likely influenced by these processes, and these α estimates probably should not be interpreted as the proportion of independently positively selected loci.

It is very feasible to examine this with simulations and their data. If it were shown that the demography and LD do not have these suspected effects, then fine. However, earlier recent research has shown that there are extensive effects of linked selection on demography, and of demography on selection estimates (e.g. Johri et al. 2020 Genet or Schrider et al. 2016 Genet and so many others). From the point of view of reader of (any) paper, it seems important to have exceptional results accompanied by thorough analysis and discussion. In this case one could hope to read about that the results are exceptional compared other findings in this (and other) species, that the data set violates assumptions of the estimation method and that thus the interpretation of proportion of loci fixed by direct selection likely is influenced and if this is found to be the case, how large is the influence of the bottleneck and likely subsequent extensive LD on the finding.

And if they find that these (seemingly rather reasonable) concerns are unjustified, then it would seem good to provide a justification for that.

The paper is very interesting and has many topics already, will add to the value if results are presented without thorough discussion?

minor comments

l. 3 ameliorate risk – is it better to say mitigate?

Reviewer #4:

Remarks to the Author:

Sorry for harking again on supplementary table 11. So, if %fixed is the percentage of simulations in which an adaptive variant arose and fixed (i.e., the percentage of simulations in which adaptation happened) then %not fixed should be the percentage of simulations in which either an adaptive variant was lost, or no adaptive variant appeared at all (i.e., adaptation did not happen).

Also, in some cases, Mean gen. to adapt is shorter than Mean gen. to 1st seg. variant. Does this mean that adaptation is more likely to fail if the adaptive variant arises later?

One minor comment: line 2 of page 15, it should read "to a new adaptive optimum".

Other than that, I think the manuscript is ready for publication.

Below we pasted the reviewer comments and responded to these in blue type. In response to the concerns of Reviewer 3, we added some clarifying statements to the manuscript as well as information about the decay of LD. In the interest of time and moving forward with the manuscript, we have completely removed the reference to alpha.

In response to Reviewer 4 we made the suggested changes and clarifications as described below.

REVIEWER COMMENTS

Reviewer #3 (Remarks to the Author):

Thanks for the extensive comments on parts of the earlier reviewer comments.

You are welcome. We hoped our responses would be helpful.

There was an issue about taking up phylogenetic species, mainly because this seemed to bring up issues outside the main focus of the ms. Describing SA and FO as highly diverged populations seems like an appropriate way to deal with the issue.

We are happy that we could agree on a solution to this.

Questions about dN/dS, DFE and alpha were brought up in the first round, more extensively in the second. In the revision, the authors have added one sentence, but mainly have not discussed the exceptional findings, or to replied to the comments on the difficulty of interpretation.

We find this summary to be a bit misleading. It sounds as if multiple reviewers were unsatisfied with our response. In fact, Reviewer 1 was apparently completely satisfied with our response to his concern. Further, it seems that since the first round, in each subsequent round only Reviewer 3 has brought up concerns about this.

It is true that the CVI lineages are exceptional in several respects. In these lineages, there are several lines of evidence for strong selection during and after colonization – from species distribution modeling and climate, from a phylogenetic signature of selection (dN/dS), and from the experimentally measured fitness differential. Each of these results support strong multivariate selection pressures. This seems to us to be a situation in which elevated functional divergence might be expected, unlike the case where the formation of a species is not accompanied by major changes in the environment. We have now added a more explicit discussion of this in the Discussion section (page26), where it is possible to discuss this in the context of the entire set of results in the paper.

As we discuss below, we are removing the alpha estimates because Reviewer 3 finds them to be so contentious (and in the interest of time) but we actually do not find them *difficult to interpret* given that they seem to fit with the other results we found for these lineages, which are in many ways different from what is commonly found for old, widespread continental populations.

To repeat, the different methods for estimating DFE and alpha have several assumptions. Here they used polyDFE, which, as others, also assumes a set of independent SNPs. Thus, it is assumed that some sites are neutral, while others undergo independently negative selection, others positive selection, as reflected in the DFE. The model allows the expected neutral and deleterious substitutions (due to drift), and then the rest are assumed to be due to positive selection. This

holds, given independence (lack of high linkage disequilibrium between loci) (and some other assumptions).(of course, in most cases these do not hold exactly).

With regard to the question of the effect of LD on the polyDFE results, we note that polyDFE actually does take distortion of the SFS due to demography, ascertainment bias, nonrandom sampling and linkage into account by incorporating nuisance parameters in the model (as in Eyre-Walker et al., 2006) as well as in its bootstrap resampling approach. This approach was tested in the Tartaru (2017, Genetics) paper using simulations under a variety of demographic scenarios.

Collections of selfing species over wide distances have moderate LD (e.g. Arabidopsis, Nordborg et al. 2002 Nature Genetics, Medicago, Branca et al. 2011 PNAS, because of considerable historical recombination). Small local populations of *A. thaliana* have high LD, which means that the sites are not independent (e.g. Nordborg et al 2002 Nature Genetics, Bomblies et al. 2010 PLoS Genet).

When there is much rapid selection over some area in the genome (as the flowering time loci here), the effects are expected to spread over large areas of the genome when effective recombination is low and LD is high, as expected to be here. So there is opportunity for strong selection, as the authors accurately state, but also large opportunity for effects due to nonindependence of loci (data are not shown).(And of course, for other demographic effects). The shape of the DFE and the surprisingly high estimates of α (60-50 %) are likely influenced by these processes, and these α estimates probably should not be interpreted as the proportion of independently positively selected loci.

A couple of new points related to LD are now raised in the current review. Please note that we never suggested that α should be interpreted as the "proportion of independently positively selected loci"; rather we explicitly mentioned that α represents the proportion of "non-synonymous substitutions [that] were driven to fixation by positive selection or by linkage to positively selected alleles" Page 33 and 34, supplementary text. Further, as we discuss below, the linkage is taken into account in the polyDFE approach.

The reviewer notes that the extent of LD tends to be greater in small, bottlenecked populations relative to their parent populations. Of course, this has been found not only for selfing species but also for humans, for example (classic studies include: Dunning et al., 2000, Abecasis et al., 2001, and Pritchard and Przeworski 2001). However, as we will discuss below, this is not the pattern expected or observed in the two CVI islands.

The Nordborg et al., 2002 paper that is mentioned shows in *A. thaliana* some expected patterns given what had previously been found in human populations. For example, it shows that LD is reduced in the global population relative to very local ones. However, in that paper there is no attempt to model the demographic histories of the populations nor are there any simulations, so it is not actually so clear what aspects of population history generate the patterns of LD. Further, the data used there include sequence data for two regions, which are actually the two genomic regions with the *strongest* prior evidence of selection (*FRI* (positive selection) and *RPM1* (balancing selection)) together with a sparse set of 163 SNP variants from across the genome.

Bomblies et al., 2010 is a fantastic paper, but it does not actually examine LD at all. But from Detlef Weigel's lab there is Cao et al., 2012, which includes a very nice comparison across populations, showing genome-wide evidence for reduced LD in the global population and variation among local populations.

Results from simulations performed previously can be useful to provide some expectations of the kinds of patterns of LD we might find in Cape Verde. In the Pritchard and Przeworski 2001 paper, they conducted simulations under an island model in which they examined LD in a single island and in two separate diverged islands combined, which represents a simple form of a highly structured population. From this, it is clear that LD in a single island decays rapidly whereas LD across a structured sample (e.g., two diverged island populations together) decays much less rapidly (Fig. 1, Pritchard and Przeworski 2001). This shows the (perhaps surprising) result that LD is not necessarily expected to be higher in an isolated population. It further shows the impact that population structure can have on LD. The effect of combining the two island populations in Pritchard and Przeworski 2001 is similar to a **partial** population bottleneck in that it generates an excess of intermediate frequency variants or haplotypes and this results in increased LD. Interestingly, Pritchard and Przeworski also show evidence that population growth actually lowers linkage disequilibrium in a population. This point was examined extensively by Rogers more recently using simulations and an analysis of human data (2014 PNAS).

So, a partial bottleneck is expected to lead to increased LD. Of course, for this to occur, it requires that variation remain in the population. If a bottleneck was so strong that it effectively removed all (or 99%) of the variation from a population, LD would not be generated at all. This is simply because there would be no genetic variation to produce LD. The strong LD that is generated under partial bottlenecks requires that genetic variation remains so that correlations can be generated.

In fact, we did not include a plot showing decay of LD for the CVI island populations in the paper, but now we realize that it might have been wise to do so to help deal with such misconceptions. What we find is actually that LD decays rapidly in the individual island populations (see below), consistent with the simulations from Pritchard and Przeworski, 2001. While this would be surprising if the colonization bottleneck has a major impact on LD, the important point again here is that the colonization bottlenecks were so strong that they removed (nearly) all standing variation from the populations. In each island population, basically no variation survives the bottleneck so there is nothing left to generate strong LD. We added this plot to the supplement (Supp. Fig. 3) with a brief mention of the result in the main text (page 5, lines 9-12).

*Figure 1. **Linkage disequilibrium (LD) decay in all three populations.** X-axis shows distance between SNPs in kbp, and the y-axis the corresponding r^2 value. Decay of LD is shown for Santo Antão (blue), Fogo (orange) and Morocco (green). Lines were smoothed with locally weighted scatterplot smoothing (LOESS) and the shaded grey area represents the 95% confidence interval.*

Again, polyDFE does actually take distortion of the SFS due to demography, ascertainment bias, nonrandom sampling and linkage into account, using nuisance parameters in the model and in its bootstrap resampling approach. This approach was tested in the Tartaru (2017, Genetics) paper using simulations under a variety of demographic scenarios.

We certainly were not trying to evade the issues that were raised previously and actually felt that we had dealt with them in the sentences we added in the first round of revisions. This is what we had written after several rounds of review:

Consistent with strong positive selection acting in CVI, the estimated proportion of fixed adaptive substitutions, alpha, was extremely high in CVI (70% in Santo Antão, 62% in Fogo) relative to the Moroccan population, where the estimated proportion is effectively zero. It should be noted that population history can impact estimates of alpha so that these may be somewhat inflated due to possible fixation of deleterious variants under rapid population expansion^{39,40}. Conversely, in Morocco, alpha may be underestimated due to recent population bottlenecks. While the limited numbers of fixed and segregating sites in the relatively young CVI lineages necessarily leads to large confidence intervals on our estimates, the results are consistent with strong positive selection after a shift to new adaptive optimum in the nascent CVI lineages.

It is very feasible to examine this with simulations and their data. If it were shown that the demography and LD do not have these suspected effects, then fine. However, earlier recent research has shown that there are extensive effects of linked selection on demography, and of demography on selection estimates (e.g. Johri et al. 2020 Genet or Schrider et al. 2016 Genet and so many others).

From the point of view of reader of (any) paper, it seems important to have exceptional results accompanied by thorough analysis and discussion. In this case one could hope to read about that the results are exceptional compared other findings in this (and other) species, that the data set violates assumptions of the estimation method and that thus the interpretation of proportion of loci fixed by direct selection likely is influenced and if this is found to be the case, how large is the influence of the bottleneck and likely subsequent extensive LD on the finding. And if they find that these (seemingly rather reasonable) concerns are unjustified, then it would seem good to provide a justification for that. The paper is very interesting and has many topics already, will add to the value if results are presented without thorough discussion?

Above we show that LD actually decays rather rapidly in the Santo Antão and Fogo populations, consistent with the complete bottleneck that occurred (in contrast to the pattern that results from a partial bottleneck). Further, the nuisance parameters in polyDFE, and the bootstrap resampling approach of Tartaru et al., already deals with issues surrounding LD and this has been tested.

In any case, as we stated in the text already (pasted above), confidence intervals on the DFE estimates are high, as expected due to the short time and rather few variants that contribute to these estimates. We believe the caveats we mentioned should have been sufficient, but in the interest of time and moving forward with this manuscript we have removed the reference to alpha.

minor comments

I. 3 ameliorate risk – is it better to say mitigate?

Thank you for this suggestion. We changed this in the text.

Reviewer #4 (Remarks to the Author):

Sorry for harking again on supplementary table 11. So, if %fixed is the percentage of simulations in which an adaptive variant arose and fixed (i.e., the percentage of simulations in which adaptation happened) then %not fixed should be the percentage of simulations in which either an adaptive variant was lost, or no adaptive variant appeared at all (i.e., adaptation did not happen).

We have changed this in Supp Table 11 so that it now reads:

% not fixed: the percentage of simulations where an adaptive variant arose but did not fix

We have also added a new column for clarity and completeness, "% no mutations", to provide information about the percent of simulations where no adaptive variant arose. "%fixed", "%not fixed" and "%no mutations" sum to 1.

Also, in some cases, Mean gen. to adapt is shorter than Mean gen. to 1st seg. variant. Does this mean that adaptation is more likely to fail if the adaptive variant arises later?

Thank you for pointing out this discrepancy. We went back to the calculations from the simulations and found that we had incorrectly included cases where no adaptive variant arose, and we had used the last generation of the simulations in this case to represent the time where the variant arose. We have rectified this issue. Now the results are logically consistent, with the mean generations to adapt longer than the mean generations to the first segregating variant.

We also reordered the columns to make the table easier to digest.

One minor comment: line 2 of page 15, it should read "to a new adaptive optimum".

Thank you. We have added 'a' here.

Other than that, I think the manuscript is ready for publication.

Thank you.

Reviewers' Comments:

Reviewer #3:

Remarks to the Author:

The review here deals with issues to be still addressed: the assessment of the role of positive selection using the genomewide data, estimating DFEs, d_{sel}/d_{neu} and α . As mentioned before, all of these methods make assumptions e.g., about independence of the sites for which analysis is performed.

The authors have now removed the α estimates but retained the d_{sel}/d_{neu} estimates and the DFE and have added a figure on LD in Morocco and in the present SA and FO populations.

An easy way to deal with the issues would be to send a short e-mail to Paola Tataru or Thomas Bataillon to ask for their comments.

As the authors want to keep these results, it would be very useful to add a little additional clarification and information to help the reader, standard information to have available. 1) for estimating the DFE for Fogo and Santo Antao, what exactly is the reference point for defining the fixed substitutions, is it the shared CV lineage and how was it constituted (around lines 197-210, 698-700, and 625-630 text could be clearer, explaining once of course is enough). 2) What is the number of sites (0-fold and 4-fold) on which the DFE and d_{sel}/d_{neu} estimates are based for the two islands (e.g., in fig. 5 b would be a good place, or text). 3) please add information on the number of sites on which the LD estimates are based.

Then still a few comments on the estimation method. The polyDFE method makes assumptions of independence of sites and population sizes. Real populations of course never fully fit the assumptions but often are close enough. For the highly selfing *A. thaliana*, the assumption of independence is much more violated than in outcrossers, because recombination is so limited that any LD generated would have considerable effects (Hedrick Genetics of populations 3rd edition p. 548 and references therein, such as Weir). Here, the bottlenecks seen in Fig. 3 would be expected to generate high LD during the early stages of the two islands. This beginning LD is what likely governs the lack of independence during selection in the bottlenecked populations (so perhaps Morocco LD would be closer than final LD, even if an underestimate).

The authors refer to the Pritchard and Przeworski (2001) paper. That paper deals with random mating populations of 5000 individuals that are differentiated, and have at most low LD within populations, but mixing of populations generates higher LD. It does not seem that the conditions of these simulations are not very relevant for the *A. thaliana* situation.

No models fit perfectly, and indeed, polyDFE and other methods make some corrections for these deviations. Likewise it true that no corrections correct everything, and the deviations in the model in the present case are rather extreme. Tataru et al. (2017) examined population size changes (see supp info

(<https://www.genetics.org/content/genetics/suppl/2017/09/25/genetics.117.300323.DC1/FileS1.pdf>) of a doubling of N_e or a reduction of N_e to half of the initial size. These effects could not be completely corrected for by the methods (p. 1113). The N_e changes that *A. thaliana* experienced here are much more extreme (as diversity is less than 2 % of that on the mainland). The effects of this kind of population sizes have not been tested (at least not by Tataru et al. 2017). Tataru et al. (2017) also tested for the effect of linkage. For this, they used 1000 fully independent fragments, which were each divided to many fully independent segments. Within the segments, they had variable numbers of sites that experienced limited recombination. This setup again is very different from *A. thaliana* situation where there was likely high LD and the high degree of selfing seriously limited recombination and decay of LD during selection (as explained above). Thus, the Tataru et al. examination of deviations clearly does not answer issues for this data set. If the authors have tested the present situation with methods suggested by Tataru et al., they do not report about it.

Reviewer #4:

Remarks to the Author:

All my comments have been addressed to my satisfaction.

REVIEWER COMMENTS

Reviewer #3 (Remarks to the Author):

The review here deals with issues to be still addressed: the assessment of the role of positive selection using the genomewide data, estimating DFEs, d_{sel}/d_{neu} and α . As mentioned before, all of these methods make assumptions e.g., about independence of the sites for which analysis is performed.

The authors have now removed the α estimates but retained the d_{sel}/d_{neu} estimates and the DFE and have added a figure on LD in Morocco and in the present SA and FO populations.

An easy way to deal with the issues would be to send a short e-mail to Paola Tataru or Thomas Bataillon to ask for their comments.

As the authors want to keep these results, it would be very useful to add a little additional clarification and information to help the reader, standard information to have available. 1) for estimating the DFE for Fogo and Santo Antao, what exactly is the reference point for defining the fixed substitutions, is it the shared CV lineage and how was it constituted (around lines 197-210, 698-700, and 625-630 text could be clearer, explaining once of course is enough).

We have clarified this at lines 206-209, pasted here for reference:

We calculated whole-genome d_{sel}/d_{neu} on the branch between Morocco and the most recent common ancestor of the two islands (i.e., variation fixed derived in CVI and absent from Morocco) as well as on the branches leading to each individual island (i.e., variation private to a single island, and fixed there) (Fig. 5a).

And at lines 629-636:

We used custom scripts (<https://github.com/HancockLab/CVI>) to compute the d_{sel}/d_{neu} ratio, defined as the rate ratio of 0-fold non-synonymous to 4-fold synonymous substitutions, scaled by the number of sites at risk for each category... ..To address the divergence branch between the two islands and the mainland, we used mutations that are fixed derived in Cape Verde and absent from Morocco. To address the branches leading to each individual island, we used mutations that are fixed derived in one island and absent from the other island and Morocco.

2) What is the number of sites (0-fold and 4-fold) on which the DFE and d_{sel}/d_{neu} estimates are based for the two islands (e.g., in fig. 5 b would be a good place, or text).

We added mention to this at lines 631-633:

Genome-wide, after discounting sites with more than 5% missing data, the number of sites at risk for 0-fold and 4-fold mutations were respectively 5967270 and 1332660.

3) please add information on the number of sites on which the LD estimates are based.

The numbers of sites were 55, 645 for Santo Antão, 56, 173 for Fogo and 1, 435, 763 for Morocco. This information has been added to the Linkage Disequilibrium section of the Supplementary Methods.

Then still a few comments on the estimation method. The polyDFE method makes assumptions of independence of sites and population sizes. Real populations of course never fully fit the assumptions but often are close enough. For the highly selfing *A. thaliana*, the assumption of independence is much more violated than in outcrossers, because recombination is so limited that any LD generated would have considerable effects (Hedrick Genetics of populations 3rd edition p. 548 and references therein, such as Weir). Here, the bottlenecks seen in Fig. 3 would be expected to generate high LD during the early stages of the two islands. This beginning LD is what likely governs the lack of independence during selection in the bottlenecked populations (so perhaps Morocco LD would be closer than final LD, even if an underestimate).

We would like to reiterate that the extreme bottlenecks that occurred during colonization in CVI eliminated variation and thus the bottleneck does not result in increased LD. This is explained also in the discussion from the previous round.

The authors refer to the Pritchard and Przeworski (2001) paper. That paper deals with random mating populations of 5000 individuals that are differentiated, and have at most low LD within populations, but mixing of populations generates higher LD. It does not seem that the conditions of these simulations are not very relevant for the *A. thaliana* situation.

Thank you for your comment. Selfing would be expected to shift the level of LD consistently across models (i.e., no reason for an interaction effect here), so the comparison of patterns between different simulation models is valid.

No models fit perfectly, and indeed, polyDFE and other methods make some corrections for these deviations. Likewise it true that no corrections correct everything, and the deviations in the model in the present case are rather extreme. Tataru et al. (2017) examined population size changes (see supp info <https://www.genetics.org/content/genetics/suppl/2017/09/25/genetics.117.300323.DC1/FILES1.pdf>) of a doubling of N_e or a reduction of N_e to half of the initial size. These effects could not be completely corrected for by the methods (p. 1113). The N_e changes that *A. thaliana* experienced here are much more extreme (as diversity is less than 2 % of that on the mainland). The effects of this kind of population sizes have not been tested (at least not by Tataru et al. 2017). Tataru et al. (2017) also tested for the effect of linkage. For this, they used 1000 fully independent fragments, which were each divided to many fully independent segments. Within the segments, they had variable numbers of sites that experienced limited recombination. This setup again is very different from *A. thaliana* situation where there was likely high LD and the high degree of selfing seriously limited recombination and decay of LD during selection (as explained above). Thus, the Tataru et al. examination of deviations clearly does not answer issues for this data set. If the authors have tested the present situation with methods suggested by Tataru et al., they do not report about it.

The reviewer mentions two caveats of the DFE inference method used here. These are 1) the impact of LD and 2) of population history (i.e., changes in N_e) on the DFE estimates. We have added mention to these caveats, together with caveats of d_{sel}/d_{neu} at lines 230-236:

It should be noted that population history can impact estimates of d_{sel}/d_{neu} so that these may be somewhat inflated due to possible fixation of deleterious variants under rapid population expansion^{41,42}. Conversely, in Morocco, d_{sel}/d_{neu} may be underestimated due to recent population bottlenecks⁴¹. It should also be noted that linkage disequilibrium and demography can violate assumptions of the DFE inference⁴⁰. However, the method used here takes these effects into account using nuisance parameters, and we find a rather rapid LD decay in each island (Supplementary Fig. 3).

Reviewer #4 (Remarks to the Author):

All my comments have been addressed to my satisfaction.